

# A Parameterization of Sub-grid Topographical Effects on Solar Radiation in the E3SM Land Model (Version 1.0): Implementation and Evaluation Over the Tibetan Plateau

Dalei Hao[1], Gautam Bisht[1], Yu Gu[2], Wei-Liang Lee[3], Kuo-Nan Liou[2] and L. Ruby Leung[1]

[1]Atmospheric Sciences and Global Change Division, Pacific Northwest National Laboratory, Richland, WA, USA
[2]Joint Institute for Regional Earth System Science and Engineering and Department of Atmospheric and Oceanic Sciences, University of California, Los Angeles, CA, USA
[3]Research Center for Environmental Changes, Academia Sinica, Taipei, Taiwan

*Correspondence to*: Dalei Hao (dalei.hao@pnnl.gov)

**Abstract.** Topography exerts significant influences on the incoming solar radiation at the land surface. A few stand-alone regional and global atmospheric models have included parameterizations for sub-grid topographic effects on solar radiation. However, nearly all Earth System Models (ESMs) that participated in the Coupled Model Intercomparison Project (CMIP6) use a plane-parallel (PP) radiative transfer scheme that assumes the terrain is flat. In this study, we incorporated a well-validated sub-grid topographic (TOP) parameterization in the Energy Exascale Earth System Model (E3SM) Land Model (ELM) version 1.0 to quantify the effects of sub-grid topography on solar radiation flux, including the shadow effects and multi-scattering between adjacent terrain. The Moderate Resolution Imaging Spectroradiometer (MODIS) data was used to evaluate the performance of ELM. We studied the role of sub-grid topography by performing ELM simulations with the PP and TOP schemes over the Tibetan Plateau (TP). Additional ELM simulations were performed at multiple spatial resolutions to investigate the role of spatial scale on sub-grid topographic effects on solar radiation. When compared to MODIS data, incorporating the sub-grid topographic effects overall reduces the biases of ELM in simulating surface energy balance, snow cover and surface temperature especially in the high-elevation and snow-cover regions over the TP. Topography has non-negligible effects on surface energy budget, snow cover, and surface temperature over the TP. The absolute differences in surface energy fluxes for net solar radiation, latent heat flux, and sensible heat flux between TOP and PP exceed 20 W/m2, 10 W/m2, and 5 W/m2, respectively. The differences in land surface albedo, snow cover fraction, and surface temperature between TOP and PP exceed 0.1, 20%, and 1K, respectively. The magnitude of the sub-grid topographic effects is dependent on seasons and elevations, and is also sensitive to the spatial scales. Although the sub-grid topographic effects on solar radiation are more significant with more spatial details at finer spatial scales, they cannot be simply neglected at coarse spatial scales. The inclusion of sub-grid topographic effects on solar radiation parameterization in ELM will contribute to advancing our understanding of the role of the surface topography on terrestrial processes over complex terrain.





## 1. Introduction

Earth system models (ESMs), which simulate the interactions between atmosphere, land, ocean and cryosphere systems, are powerful tools for understanding, reconstructing and projecting the Earth's climate (Bonan and Doney, 2018). Land surface
models (LSMs) in ESMs represent the terrestrial water, energy, and carbon cycles (Dickinson et al., 2006). However, most of the state-of-the-art LSMs necessarily adopt some oversimplified and unrealistic schemes to treat the transfer of radiation, heat, water and carbon. For example, lateral transport of water and energy in the subsurface and sub-grid topographic effects on solar radiation are neglected (Fan et al., 2019). These simplifications could lead to large uncertainties especially at finer spatial scales (Fisher and Koven, 2020; Prentice et al., 2015).

The Energy Exascale Earth System Model (E3SM), a relatively new fully coupled ESM supported by the U.S. Department of Energy (DOE), aims to tackle the grand challenge of actionable predictions of Earth system variability and change (Leung et al., 2020; Golaz et al. 2019). With the capabilities to run at relatively high resolution (Caldwell et al., 2019) and including more realistic human-natural processes (Zhou et al., 2020), E3SM provides a good opportunity to better understand the complex earth system processes and their interactions. However, improving the representations of the complex, multi-scale
processes in the earth system is important to more fully realize the benefits of high-resolution modeling.

As the horizontal grid spacing of ESMs increases, topography is expected to exert more significant influences on many land surface processes including surface energy balance, surface hydrology, and snowmelt. The incoming and reflected solar radiations, as well as their direct and diffuse components, depend on surface topography (Dubayah and Rich, 1995; Hao et al., 2019a, 2019b). Topography modifies direct radiation reaching the Earth surface through self-shadowing or blocking by
adjacent topography. Topography also decreases diffuse radiation from sky by decreasing the portion of the visible sky and increases reflected radiation from adjacent topography due to the multi-scattering effects (Dubayah, 1992; Proy et al., 1989). The changes in net solar radiation due to topography significantly influence surface energy budget (Gu et al., 2012; Lee et al., 2019; Liou et al., 2007), surface hydrology (Lee et al., 2015; Zhang et al., 2018), snowmelt (Zaramella et al., 2018), precipitation (Gu et al., 2020), and vegetation distribution (Alexander et al., 2016). Incorporating the sub-grid topographic
effects on solar radiation into LSMs such as the E3SM Land Model (ELM) is key to enhancing our understanding and modeling of surface processes and land-atmosphere interactions in regions of complex terrain, with potential remote effects through excitation of Rossby waves in the atmosphere (Koster et al., 2016).

However, nearly all ESMs (including E3SM) that participated in the Coupled Model Intercomparison Project Phase 6 (CMIP6) neglect the sub-grid topographic effects on solar radiation. Sub-grid topographic effects have been recognized and
parameterized in a few regional weather and climate models (Arthur et al., 2018; Gu et al., 2020) and global climate models (Lee et al., 2015). However, most CMIP6-class ESMs adopt simple plane-parallel (PP) radiative transfer schemes based on a two-stream approximation, which assumes that topography is flat (Dai et al., 2004; Dickinson, 1983; Sellers, 1985). Such simplified radiation parameterizations do not account for sub-grid topographic effects and can lead to large systematic biases in simulating land surface processes over complex terrain (Fan et al., 2019; Lee et al., 2019; Song et al., 2020). Song et al.
(2020) reported that both CLM4.5 and CLM5.0 failed to capture the asymmetric diurnal cycles of solar radiation, surface albedo and carbon fluxes in a mountainous rainforest in Costa Rica. Lee et al. (2019) showed that accounting for the sub-grid topographic effects in the Community Land Model (CLM)-4.0 reduced the biases of reflected solar radiation over the Tibetan Plateau (TP).

Sub-grid topographic parameterizations in the LSMs need to account for the effects of sub-grid topography without
significantly increasing the computational cost. Sub-grid radiation fluxes can be explicitly calculated using a high-resolution digital elevation model (DEM) and then averaged to derive grid-scale radiation fluxes (Dubayah, 1992). However, this approach involves a vast data processing and thus introduces substantial computational burden (Helbig and Löwe, 2012). Parameterizations for sub-grid topography based on the statistical characteristics of sub-grid topography (Dubayah, 1990; Essery and Marks, 2007; Gu et al., 2020; Helbig and Löwe, 2012; Lee et al., 2011; Müller and Scherer, 2005) provide a
computationally efficient approach for LSMs. Lee et al. (2011) used 3D Monte Carlo photon tracing simulations to develop a parameterization scheme where a set of multiple linear regression equations associate the sub-grid topographic effects on





solar radiation with the domain-averaged topographic factors. The parameterization scheme developed by Lee et al. (2011) is computationally efficient because the domain-averaged topographic factors can be calculated a priori based on high-resolution DEM. This parameterization has been successfully applied in the Weather Research and Forecasting (WRF) model (Gu et al., 2012; Liou et al., 2013), CLM4.0 (Lee et al., 2015, 2019), and Taiwan Earth System Model Version 1 (TaiESM) (Lee et al., 2020).

The objective of this study is to update and evaluate the radiative transfer scheme to account for sub-grid topographic effects on solar radiation in ELM. We implemented the computationally efficient and physically realistic sub-grid parameterization scheme for solar radiation of Lee et al. (2011) into ELM. ELM simulations over the TP were performed with and without the sub-grid topographic parameterization for 2000-2010. A suite of remotely sensed data from the Moderate Resolution Imaging Spectroradiometer (MODIS) were used to evaluate ELM simulated surface energy balance, snow cover and surface temperature for different seasons. The sensitivity of the sub-grid topographic effects to spatial scales was tested by performing ELM simulations at multiple spatial resolutions.

## 2. Materials and methods

### 2.1. Model overview

ELM (Version 1.0) is based on the Community Land Model Version 4.5 (CLM4.5) (Golaz et al., 2019). ELM uses the two-stream approximation methods to calculate canopy radiation flux and accounts for the radiative effects of black carbon and dust on snow (Dang et al., 2019). New features in ELM to better represent land surface processes include an updated representation of soil hydrology, improved treatment of ecosystem carbon dynamics, a novel topography-based sub-grid spatial structure, and an irrigation scheme constrained by water management (Bisht et al., 2018; Tang and Riley, 2018; Tesfa and Leung, 2017; Zhou et al., 2020).

### 2.2. Sub-grid topographic parameterization

For a flat surface, the incoming solar radiation is composed of direct radiation ($F_{dir}^{PP}$) from sun, diffuse radiation ($F_{dif}^{PP}$) from sky and multi-scattering radiation ($F_{couple}^{PP}$). In ELM-v1.0, $F_{dir}^{PP}$ and $F_{dif}^{PP}$ over flat surfaces are derived by the two-stream approximations (Oleson et al, 2013). In contrast, the solar radiation parametrization of Lee et al., (2011) divides the incoming solar radiation over mountains into five components: 1) direct flux ($F_{dir}^{TOP}$) represents photons that are transmitted from the sun to the ground surface without encountering any reflection or scattering; 2) the direct-reflected flux ($F_{rdir}^{TOP}$) represents photons that are not scattered photons (i.e., $F_{dir}^{TOP}$) reflected by surrounding terrain; 3) diffuse flux ($F_{dif}^{TOP}$) represents photons that are scattered by atmospheric particles, but are not reflected by the ground surface; 4) diffuse-reflected flux ($F_{rdif}^{TOP}$) represents scattered photons (i.e., $F_{dif}^{TOP}$) reflected by surrounding terrain; and 5) coupled flux ($F_{couple}^{TOP}$) represents remaining photons that are reflected multiple times or scattered by ground surface and atmospheric particles. Lee et al. (2011) used the radiation fluxes over flat surfaces to calculate the radiation fluxes over mountainous terrain using sub-grid topographic factors. The relative deviation ($f_{dir}$) of direct flux between flat surface and mountain under the same atmospheric condition is defined as:

$$f_{dir} = \frac{F_{dir}^{TOP} - F_{dir}^{PP}}{F_{dir}^{PP}} \qquad (1)$$

The relative deviation ($f_{rdir}$) of direct-reflected flux over mountains to direct flux over flat surfaces is defined as:





$$f_{rdir} = \frac{F_{rdir}^{TOP}}{F_{dir}^{PP}} \qquad (2)$$

Similarly, the relative deviations ($f_{dif}$ and $f_{rdif}$) of diffuse and diffuse-reflected fluxes are expressed as:


$$f_{dif} = \frac{F_{dif}^{TOP} - F_{dif}^{PP}}{F_{dif}^{PP}} \qquad (3)$$

$$f_{rdif} = \frac{F_{rdif}^{TOP}}{F_{dif}^{PP}} \qquad (4)$$

In theory, these four relative deviations (i.e., $f_{dir}, f_{rdir}, f_{dif}$ and $f_{rdif}$) depend on solar illumination geometry and sub-grid topographic distribution. Based on a series of 3D Monte Carlo photon tracing simulations, Lee et al. (2011) built
a multiple linear regression parameterization to well predict these four relative deviations. The parameterization of Lee et al. (2011) uses four variables that include the standard deviation of elevation ($\sigma_h$) within a gridcell, grid averaged values of cosine of the local solar incident angle ($\bar{\mu}$), sky view factor ($\overline{V_d}$) and terrain configuration factor ($\overline{C_T}$). Lee et al. (2011) parameterization is given as:

$$\left[ f_{dir}\, f_{rdif}\, f_{rdir}\, f_{rdif} \right]^T = A \cdot \left[ \bar{\mu}\; \sigma_h\; \overline{V_d}\; \overline{C_T}\; 1 \right]^T \qquad (5)$$

where $A$ represents the fitting parameter matrix, which was obtained based on the data generated by the 3D Monte Carlo simulations. The sky view factor ($V_d$) represents the portion of visible sky limited by surrounding terrain (Zakšek et al., 2011), while the terrain configuration factor ($C_T$), the counterpart of the sky view factor, represents the portion of surrounding terrain which is visible to the ground target (Dozier and Frew, 1990). For an unobstructed infinite slope with the slope of $\alpha$ and aspect of $\beta$ and a given solar illumination geometry (i.e., solar zenith angle (SZA) and solar azimuth angle
(SAA)), the cosine of the local solar incident angle ($\mu$) can be calculated by:

$$\mu = \cos(SZA) \cdot \cos(\alpha) + \sin(SZA) \cdot \sin(\alpha) \cdot \cos(SAA - \beta) \qquad (6)$$

The SZA and SAA are assumed to be constant within a gridcell, but $\alpha$ and $\beta$ vary within a gridcell. The gridcell average solar incident angle, $\bar{\mu}$, can be expressed as:

$$\bar{\mu} = \overline{\cos(SZA) \cdot \cos(\alpha)} + \overline{\sin(SZA) \cdot \sin(\alpha) \cdot \cos(SAA - \beta)}$$
$$= \cos(SZA) \cdot \overline{\cos(\alpha)} + \sin(SZA) \cdot \cos(SAA) \cdot \overline{\sin(\alpha) \cdot \cos(\beta)}$$
$$+ \sin(SZA) \cdot \sin(SAA) \cdot \overline{\sin(\alpha) \cdot \sin(\beta)} \qquad (7)$$

where overlines represent grid averaged values. To further improve the regression parameterization, $\mu$, $V_d$ and $C_T$ are normalized by $\cos(\alpha)$. The land surface albedo is adjusted, instead of modifying incoming solar radiation, to maintain energy conservation between the atmospheric and land model (Lee et al., 2015). Specifically, to keep the absorbed solar
radiation of the ground surface unchanged, Lee et al. (2015) built the relationship between direct ($\alpha_{dir}^{TOP}$) and diffuse ($\alpha_{dif}^{TOP}$) albedo over mountains and those ($\alpha_{dir}^{PP}$ and $\alpha_{dif}^{PP}$) over flat surfaces as:

$$F_{dir}^{PP} \cdot (1 - \alpha_{dir}^{TOP}) = (F_{dir}^{TOP} + F_{rdir}^{TOP}) \cdot (1 - \alpha_{dir}^{PP}) \qquad (8)$$



$$F_{dif}^{PP} \cdot (1 - \alpha_{dif}^{TOP}) = (F_{dif}^{TOP} + F_{rdif}^{TOP}) \cdot (1 - \alpha_{dif}^{PP}) \qquad (9)$$

Substituting equations (1-4) into equations (8-9) leads to

$$\alpha_{dir}^{TOP} = 1 - (1 + f_{dir} + f_{rdir}) \cdot (1 - \alpha_{dir}^{PP}) \qquad (10)$$

$$\alpha_{dif}^{TOP} = 1 - (1 + f_{dif} + f_{rdif}) \cdot (1 - \alpha_{dif}^{PP}) \qquad (11)$$

The parameterizations represented by equations (5, 10-11) were implemented in ELM to account for the sub-grid topographic effects on solar radiation fluxes. The fitting parameter matrix was pre-calculated using high resolution DEM (see Section 2.4).

**2.3. Model setup and experiment design**

TP, known as the Third Pole, plays an important role in regulating the earth climate system (Lu et al., 2020; Yang et al., 2009). TP has complex topographic features, where the central part is relatively flat, and the western and southern regions have remarkable terrain undulations (Figure 1). Figure S1 shows the heterogeneous spatial variations of the topographic factors used in the solar radiation parameterizations. Therefore, TP is an ideal region to study topography-related land surface processes in ELM.

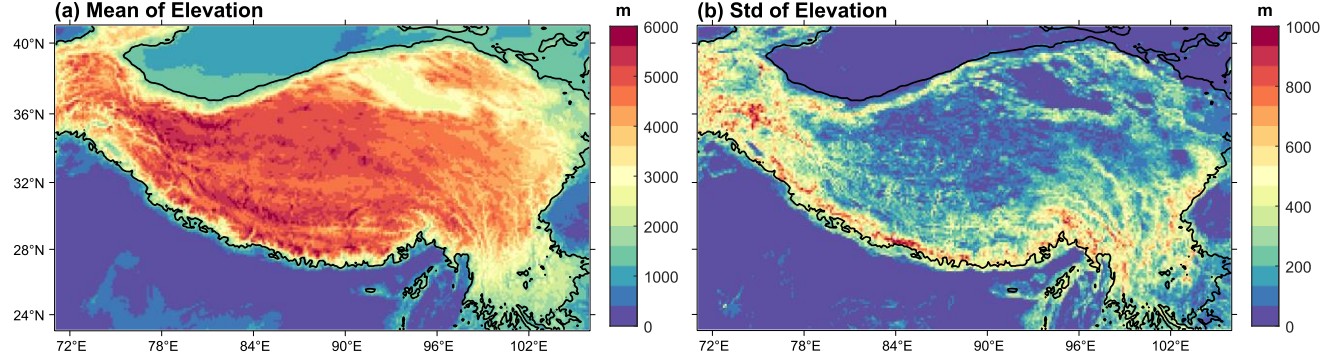

**Figure 1.** Geographical distributions of the a) mean and (b) standard deviations of elevation derived from 90m DEM at 0.125°x0.125° spatial resolution over the TP. The black line represents the contour line of 1.5 km.

Offline ELM simulations over the TP were performed for the period of 2000 to 2010 with and without the topographic parameterization, and the simulations are denoted as TOP and PP, respectively. The simulations were performed in the prescribed satellite vegetation phenology mode in which seasonally varying leaf area index is prescribed based on satellite data. The Global Soil Wetness Project meteorological forcing data set version 1 (GSWP3v1) (Dirmeyer et al., 2006; Yoshimura and Kanamitsu, 2013) was used to drive the model simulations. ELM was configured to run over the TP at five different spatial resolutions including r0125 (0.125°), r025 (0.25°), r05 (0.5°), f09 (about 1°) and f19 (about 2°). The model outputs were archived at half-hourly frequency. The impact of initial conditions on subsequent analysis was avoided by discarding the results of the first year.





## 2.4. Remote sensing data

The Shuttle Radar Topography DEM (SRTM) data at 90m spatial resolution was used to derive the topographic factors required for the TOP simulations. The spatial mean and standard deviations of elevation, slope, aspect, sky view factor, and terrain configuration factor were computed for each ELM gridcell at all five spatial resolutions.

The MODIS data from 2001-2010 was used to evaluate the performance of ELM. All MODIS data listed in Table 1 were downloaded from the Google Earth Engine Platform (Gorelick et al., 2017). Specifically, these data included both direct (i.e.,
black-sky) and diffuse (i.e., white-sky) albedo data from the daily MCD43A3 v6 products with 500m spatial resolution (Schaaf et al., 2002); snow cover data from daily MOD10A1 v6 products at 500m spatial resolution (Hall et al., 2002); both daytime and nighttime surface temperature data from the daily MOD11A1 v6 products with 1km spatial resolution (Wan, 2014); and latent heat flux data from the 8-day MOD16A2 v6 products with 500m spatial resolution (Mu et al., 2007, 2011). Only the MODIS pixels with good quality indicated by the Quality Assurance flag were used in the analysis. All MODIS
data were upscaled using the area-weighted averaging method to conform with the ELM resolutions.

**Table 1.** Specifications of the remote sensing data used in the study.

| Parameters | Product Names | Spatial resolution | Temporal resolution | Periods | References |
|---|---|---|---|---|---|
| Land surface albedo | MCD43A3.006 | 500m | daily | 2000.02-2010.12 | (Schaaf et al., 2002) |
| Snow cover | MOD10A1.006 | 500m | daily | 2000.02-2010.12 | (Hall et al., 2002) |
| Surface temperature | MOD11A1.006 | 1km | daily | 2000.03-2010.12 | (Wan, 2014) |
| Latent heat flux | MOD16A2.006 | 500m | 8-day | 2001.01-2010.12 | (Mu et al., 2007, 2011) |
| DEM | SRTM | 90m | - | - | (Jarvis et al., 2008) |

## 2.5. Model Evaluation

MODIS data (introduced in Section 2.4) was used to evaluate the performance of both TOP and PP at r0125 resolution. All MODIS data from 2001-2010 was averaged to the seasonal scales. The MODIS instantaneous surface diffuse and direct albedo datasets were derived for the local solar noon, while the MODIS instantaneous surface temperature data was derived for daytime and nighttime corresponding to 10:30 and 22:30 local solar time, respectively. The ELM simulated surface albedo and surface temperature were extracted at the corresponding MODIS time to compute the seasonally-averaged values.
ELM simulations were evaluated by computing the difference between MODIS data and PP ($\delta_{PP}$) and TOP ($\delta_{TOP}$). Furthermore, the relative difference between PP and TOP with respect to the MODIS data was computed as $|\delta_{TOP}|-|\delta_{PP}|$.

## 2.6. Model Analysis

The ELM-based simulations, TOP and PP, at r0125 resolution were used to analyze the sub-grid topographic effects on surface energy budget (i.e., land surface albedo, net solar radiation, sensible heat flux and latent heat flux), snow cover and
surface temperature. Surface temperature was calculated from the emitted longwave radiation using the Stefan-Boltzmann law, with the assumption that surface emissivity is equal to 1. The seasonally-averaged values were computed from half-hourly ELM outputs for different seasons: winter (DJF), spring (MAM), summer (JJA) and autumn (SON). Both the absolute differences (i.e., TOP-PP) and relative differences (i.e., (TOP-PP)/PP) were used to analyze the sub-grid topographic effects as well as their spatial patterns.





The relationship between sub-grid topographic effects and elevations was analyzed by dividing the elevations into four intervals: 1.5-2.5 km, 2.5-3.5 km, 3.5-4.5 km and >4.5 km, which account for about 11%, 9%, 14% and 23% of the rectangular region shown in Figure 1, respectively. Regions with elevations lower than 1.5 km were not included in the analysis due to their flat topography (Figure 1). Gridcells with a mean slope of zero were also excluded from this analysis. Additionally, gridcells with zero snow cover fraction were excluded when analyzing results for snow cover.

A random forest model was used to analyze the sensitivity of the topography-driven differences to the topographic factors. The random forest model is a regression tree-based bootstrapped non-parametric machine learning model, which allows the calculations of the variable importance (Breiman, 2001). Specifically, based on equations 5 and 7, we selected the quantities $\overline{\sin(\alpha) \cdot \cos(\beta)}$, $\overline{\sin(\alpha) \cdot \sin(\beta)}$, $\sigma_h$, $\overline{V_d}$, $\overline{C_T}$, and the PP simulated land surface albedo as driving variables in the random forest model to understand their relative importance in controlling the sub-grid topographic effects simulated by the sub-grid
parameterization.

In order to study the sensitivity of the sub-grid topographic effects to spatial scales, ELM outputs for TOP and PP at the remaining four spatial resolutions (r025 to f19) were processed to derive seasonally-averaged values.

## 3. Results

### 3.1. Comparison with MODIS data

Overall, TOP shows better consistencies with the MODIS land surface albedo data than PP (Figures 2-3). In the western regions, PP overestimates direct and diffuse albedo in winter, and underestimates them in spring (Figures 2-3), possibly due to the bias of snow cover in the model simulations (Figure 4). In most other regions, PP generally overestimates direct albedo for different seasons, and underestimates diffuse albedo except in summer. The bias in PP, $\delta_{PP}$, for direct and diffuse albedo can exceed 0.2. Compared to PP, direct albedo of TOP overall has smaller bias relative to the MODIS data in the
western regions, except in spring (Figure 2). The improvement of TOP in direct albedo can be larger than 0.1. However, for diffuse albedo, the performance of TOP in most regions is similar to or even worse than PP (Figure 3). The difference of diffuse albedo between $\delta_{TOP}$ and $\delta_{PP}$ is within 0.02 in about 86% of the whole rectangular regions in Figure 3, in winter.

TOP generally outperforms PP in winter, when compared to the MODIS snow cover, surface temperature, latent heat flux data (Figures 4-7). In the western regions, PP has higher snow cover fractions than MODIS data in winter, but lower snow
cover fractions in other seasons. In other regions, PP has lower snow cover fractions in different seasons (Figure 4). TOP has smaller biases relative to the MODIS data than PP in winter and the absolute value of $|\delta_{TOP}|-|\delta_{PP}|$ can be larger than 10%. TOP has slightly larger biases in spring but there is no significant difference between TOP and PP in summer and autumn, due to the low snow cover. The spatial distribution of $\delta_{PP}$ in snow cover fraction is consistent with the pattern of biases in land surface albedo shown in Figures 2-3. For daytime surface temperature (Figure 5), there is a larger difference between
PP and MODIS, which can exceed 5K. TOP can reduce the biases by ~0.5-1K in the central regions, especially in winter. For nighttime surface temperature (Figure 6), PP has systematically higher values than the MODIS data but the difference between TOP and PP is not significant. For latent heat flux (Figure 7), there are big differences between PP and the MODIS data. In contrast, TOP has a slightly better performance than PP in winter, but for other seasons, TOP and PP have similar performance, when compared to the MODIS data.

As the elevation increases, TOP shows higher consistencies with the MODIS data in winter (Figure 8). When the elevation is below 3.5 km, TOP and PP have similar performance, but at higher elevation TOP overall has lower biases in direct albedo (Figure 8a), snow cover fraction (Figure 8c), daytime surface temperature (Figure 8d) and latent heat flux (Figure 8f). The bias in direct albedo is smaller in TOP as compared to PP for 54% and 63% of the study region in elevation bands 3.5-4.5 km and >4.5km, respectively. The difference in the bias for snow cover fraction between TOP and PP remains unchanged for
the elevation bands 3.5-4.5 km and > 4.5km. TOP has smaller bias in daytime surface temperature as compared to PP for 57% of the study region at elevation >4.5km. The bias in latent heat flux is smaller for TOP than PP for elevation band 3.5-





4.5km and >4.5km for ~60% of the study region. The differences in bias between PP and TOP are not significant for diffuse albedo and nighttime surface temperature in most of the regions. For example, when the elevation is above 4.5 km, the difference in biases of diffuse albedo is within 0.01 for 73% of the regions, and is within 0.02 for 91% of the regions. For nighttime surface temperature, the difference in biases is within 0.1 K in about 61% of the regions.

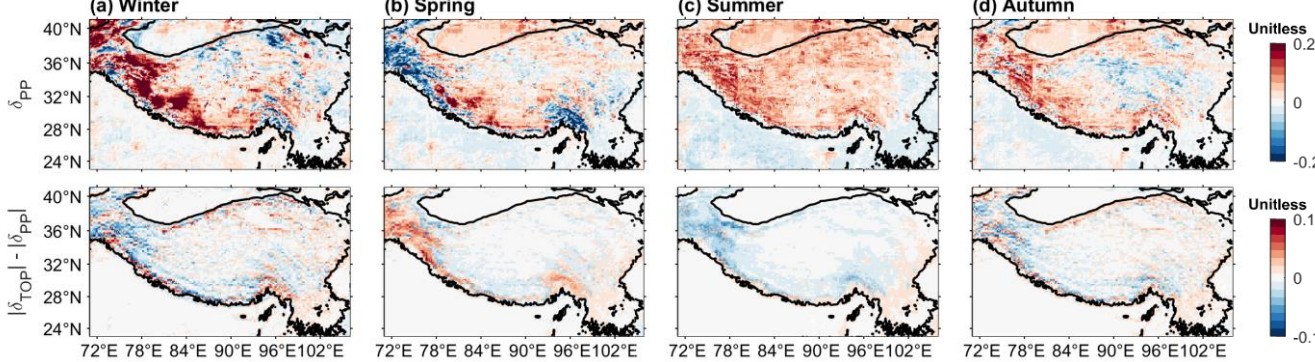

**Figure 2.** Difference ($|\boldsymbol{\delta}_{PP}|$) between PP simulated and MODIS direct albedo (top row) and the relative difference ($|\boldsymbol{\delta}_{TOP}|-|\boldsymbol{\delta}_{PP}|$) between PP and TOP with respect to MODIS data (bottom row) for different seasons.

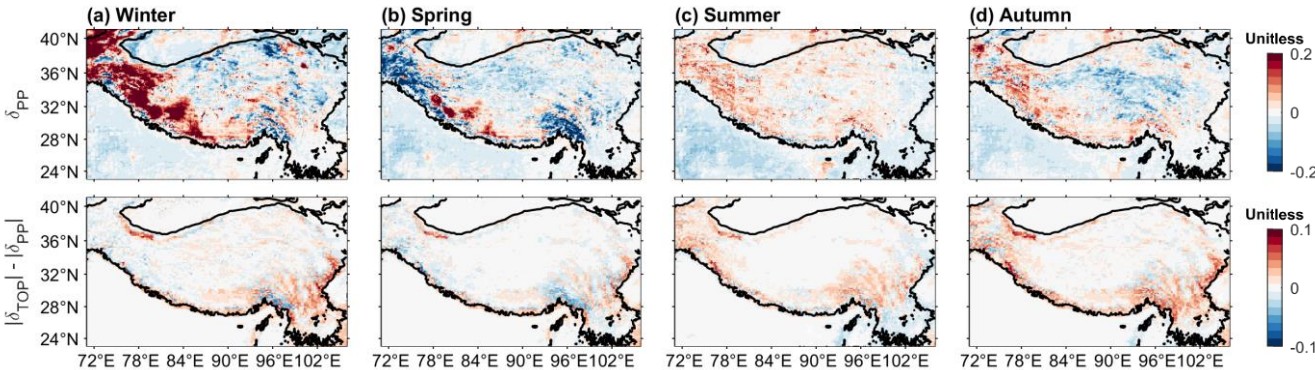

**Figure 3.** Same as Figure 2 except for diffuse albedo.

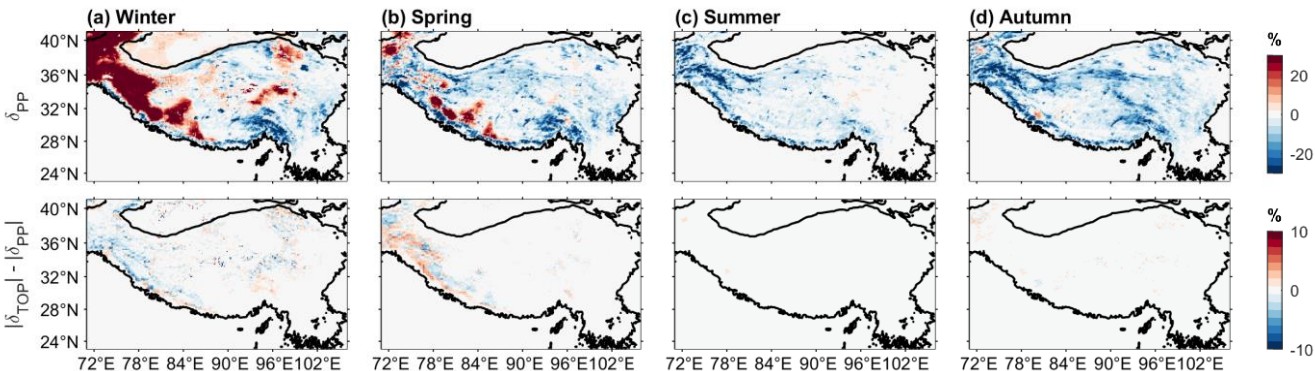

**Figure 4.** Same as Figure 2 except for snow cover fraction.





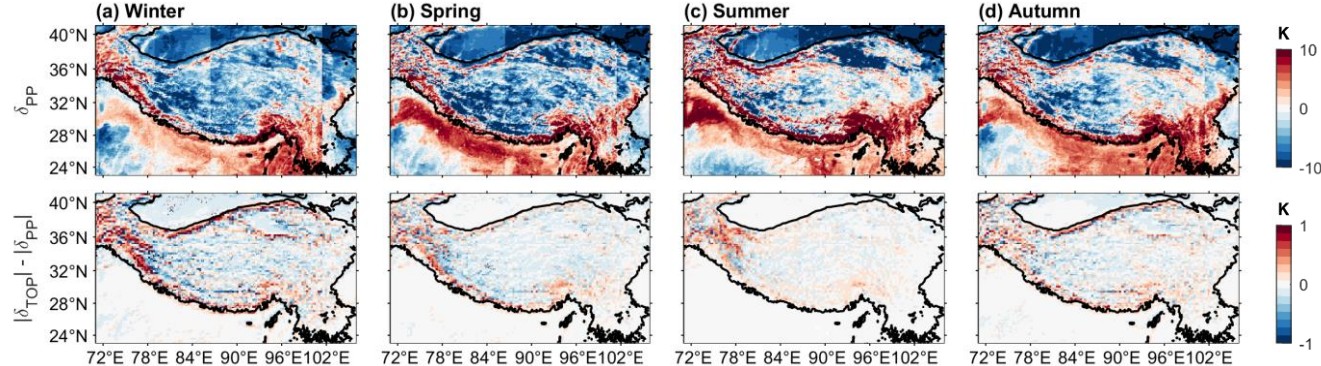

**Figure 5.** Same as Figure 2 except for daytime surface temperature.

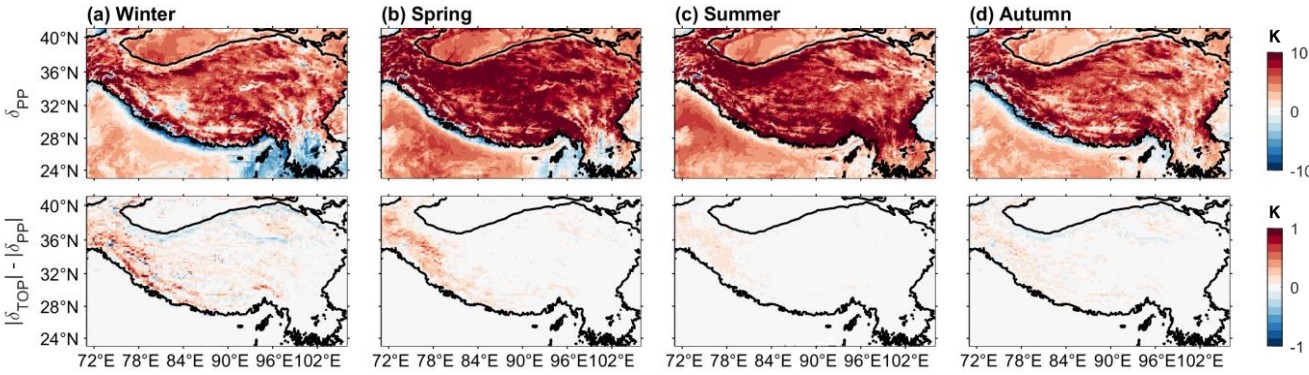

**Figure 6.** Same as Figure 2 except for nighttime surface temperature.

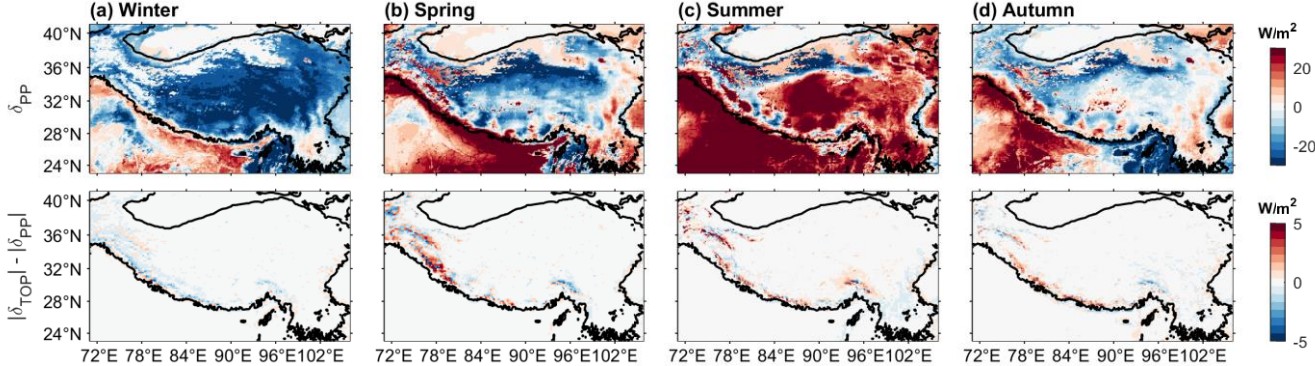

**Figure 7.** Same as Figure 2 except for latent heat flux.





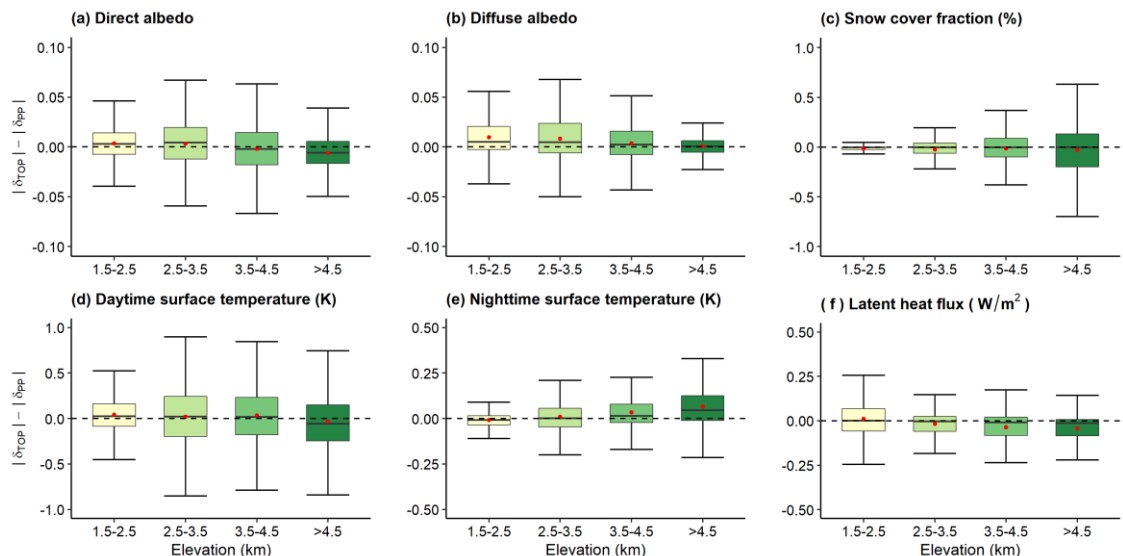

**Figure 8.** Boxplots of the differences in bias for TOP and PP ($|\boldsymbol{\delta}_{\text{TOP}}|$-$|\boldsymbol{\delta}_{\text{PP}}|$) with respect to MODIS data for (a) direct albedo, (b) diffuse albedo, (c) snow cover fraction, (d) daytime surface temperature, (e) nighttime surface temperature, and (f) latent heat flux in winter at four different elevations bands. Red points represent the mean values.

### 3.2. Sub-grid topographic effects on surface energy budget, surface temperature and snow cover

270 Compared to PP, TOP overall has higher net solar radiation (Figure 9) and lower land surface albedo (Figure 10) for different seasons. The net solar radiation for PP shows an expected and opposite spatial pattern to the land surface albedo. The absolute differences in net solar radiation between TOP and PP are as large as around 20 W/m² for different seasons, while the relative differences in the winter season are as large as 25%. In some small portions of the northern regions, TOP also shows lower net solar radiation than PP in winter and autumn, possibly due to the self-shadow or cast-shadow from the surrounding terrain. For PP, the spatial differences in surface albedo between the northwest and southeast of the study region (Figures 10a and 10b) are caused by the spatial differences in snow cover (Figures 11a and 11b). In summer, the land surface albedo in the western regions decreases due to snow melt. The land surface albedo for different seasons in the western and southern regions shows significant absolute and relative differences between TOP and PP that are as large as 0.1 and 50%, respectively, during winter. The spatial pattern of the difference in land surface albedo between TOP and PP is similar to the 280 spatial pattern of the standard deviations of elevation (Figure 1b).

Larger net solar radiation in TOP compared to PP leads to lower snow cover and higher surface temperature (Figures 11-12). TOP has lower snow cover fractions for most western regions in winter and spring (Figure 11). The absolute and relative differences of snow cover fraction between TOP and PP can be larger than -10% and -20%. Snow albedo feedback may have contributed to the large differences between TOP and PP, as larger net solar radiation in TOP reduces snow cover, which 285 may further increase the net solar radiation. Surface temperature has a similar spatiotemporal pattern as the net solar radiation (Figure 12). The absolute difference of surface temperature between TOP and PP is generally within 1 K for different seasons. The western regions have large differences in surface temperature and snow cover during winter.

TOP has higher sensible and latent heat fluxes than PP, due to the higher net solar radiation (Figures 13-14). TOP shows higher sensible heat flux than PP for different seasons, and the absolute and relative differences can be as large as 10 W/m² 290 and 20%, respectively (Figure 13). The difference in the latent heat flux is smaller compared to the difference in the sensible heat flux (Figure 14) and is generally within 5 W/m². But the relative difference in sensible heat flux may be larger than 20% in winter. How the partitioning of surface heat flux between sensible and latent heat fluxes responds to the difference in net





solar radiation between TOP and PP may vary by seasons and regions depending on the soil moisture, vegetation, and other factors.

The differences of surface energy budget, surface temperature and snow cover between TOP and PP show elevation-dependent patterns in winter (Figure 15). Generally, as the elevation increases, TOP has a lower land surface albedo and snow cover fraction than PP, but a higher net solar radiation, surface temperature and sensible and latent heat flux than PP. Taking land surface albedo as an example (Figure 15b), for elevation between 1.5-2.5 km, TOP has smaller values than PP in 53% of the regions; for elevation between 2.5 and 3.5 km, the proportion is 57%; for elevation between 3.5 and 4.5 km, the
proportion is 68%; and for elevation above 4.5 km, the proportion is 74%. With the increase of elevation bands, the larger decrease in land surface albedo of TOP leads to larger increase in surface fluxes (Figure 15a, e-f) and surface temperature (Figure 15d), along with larger decrease in snow cover (Figure 15c). In addition, the quantiles in Figure 15 also show that as the elevation increases, the relative differences of net solar radiation, snow cover fraction, surface temperature and sensible and latent heat flux between TOP and PP can become larger, except that the relative differences of land surface albedo can
exceed ±0.1 for all elevation bands.

The random forest model can well predict the sub-grid topographic effects on solar radiation with high coefficients of determination ($R^2$) for different seasons (Figure 16a-d), which demonstrates that the topographic factors can well explain the difference between TOP and PP in land surface albedo. Further sensitivity analysis (Figure 16e-h) show that the contributions of different topographic factors to the sub-grid topographic effects are different. The first two terms (i.e.,
$\overline{\sin(\alpha) \cdot \cos(\beta)}$ and $\overline{\sin(\alpha) \cdot \sin(\beta)}$), related to the subgrid distribution of slope and aspect, can account for 62.5% of the differences in surface albedo during winter (Figure 16e). The slope and aspect affect the direct solar radiation, which dominates the total solar radiation under clear-sky conditions. The sky view factor, terrain configuration factor and land surface albedo for PP, which mainly affect the diffuse and reflected radiation, account for 2.7%, 2.3% and 24.7% in winter, respectively. The dominant factors for the differences between TOP and PP can be different in different seasons (Figure 16e-
h). In summer, the contributions of the first two terms decrease to 47.1% (Figure 16g). This is because the solar position (i.e., solar illumination geometry) is different in different seasons. In winter, the solar zenith angle is large over the TP and thus there are strong shadowing effects, while the sun is moving northward and getting closer to the nadir position from spring to summer, which can lead to the reduced shadowing effects. Similar results were obtained for other variables (e.g., net solar radiation and surface temperature) and thus are not shown in this paper.




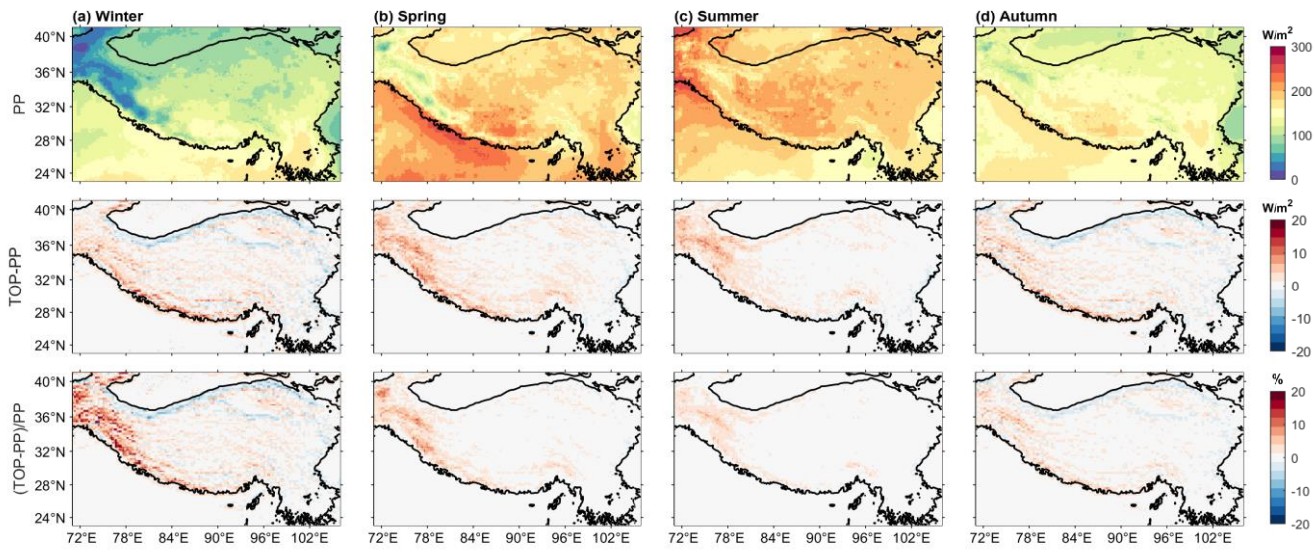

**Figure 9.** PP simulated net solar radiation for different seasons (top row). Absolute (middle row) and relative (bottom row) differences between TOP and PP for different seasons.


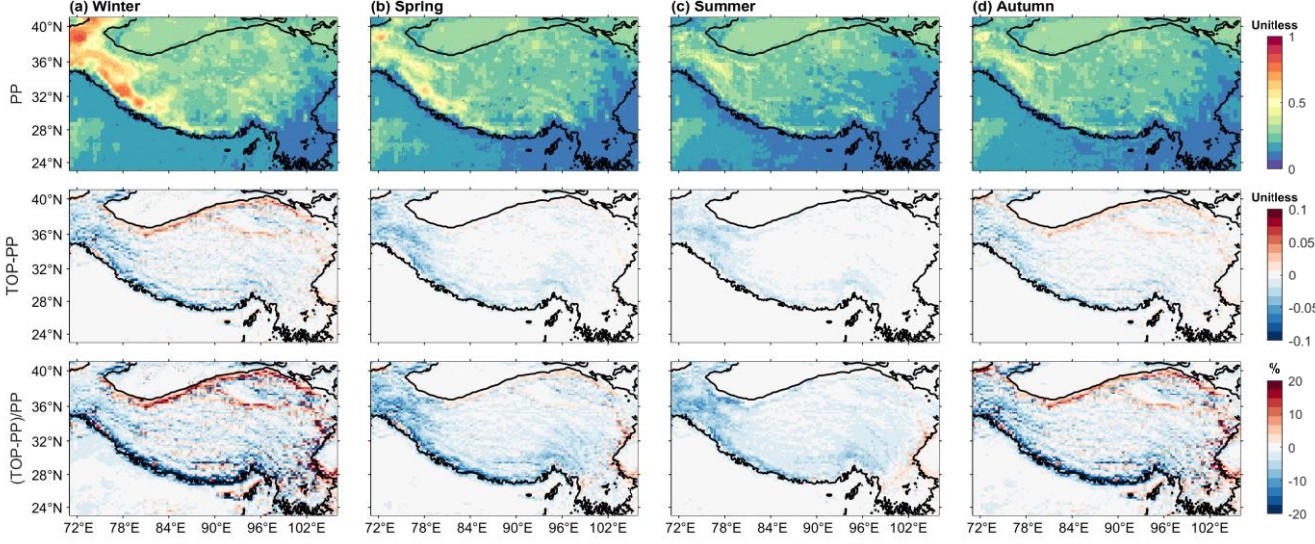

**Figure 10.** Same as Figure 9 except for land surface albedo.


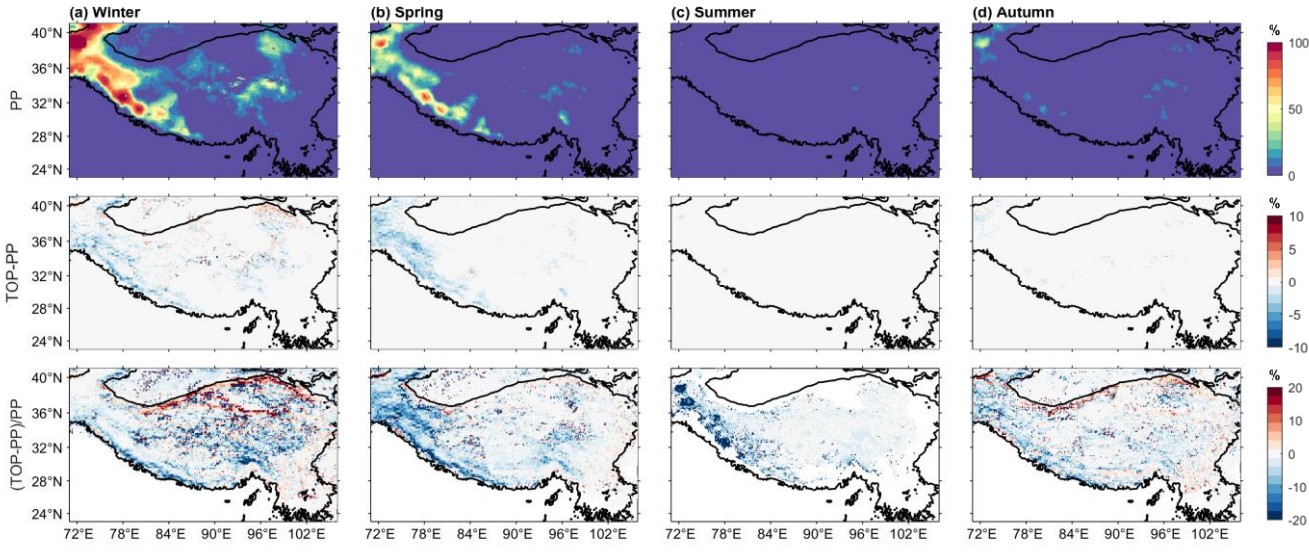

**Figure 11.** Same as Figure 9 except for snow cover fraction.

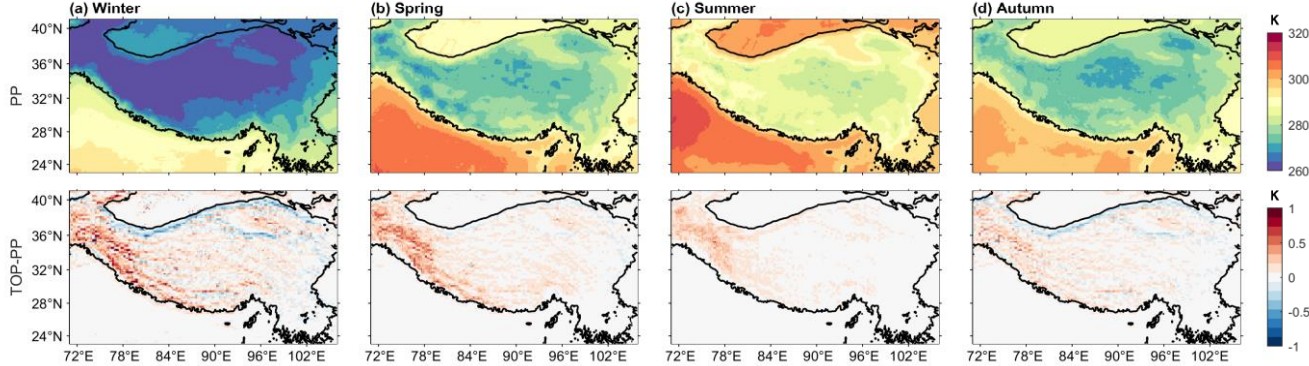

**Figure 12.** PP simulated surface temperature for different seasons (top row) and absolute differences between TOP and PP (bottom row).



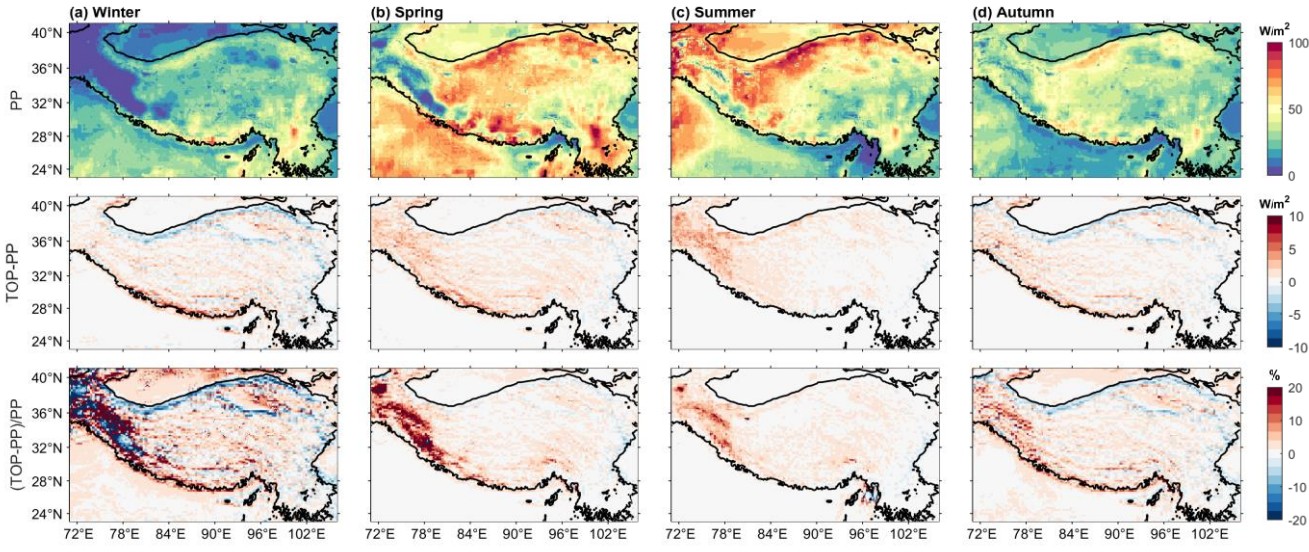

**Figure 13.** Same as Figure 9 except for sensible heat flux.

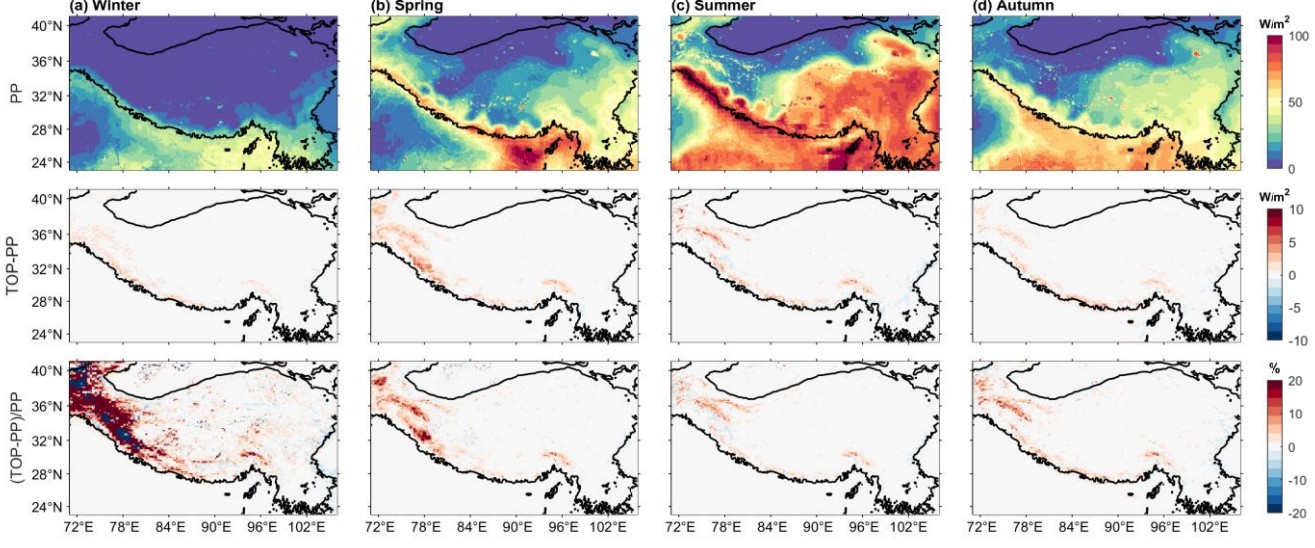


**Figure 14.** Same as Figure 9 except for latent heat flux.



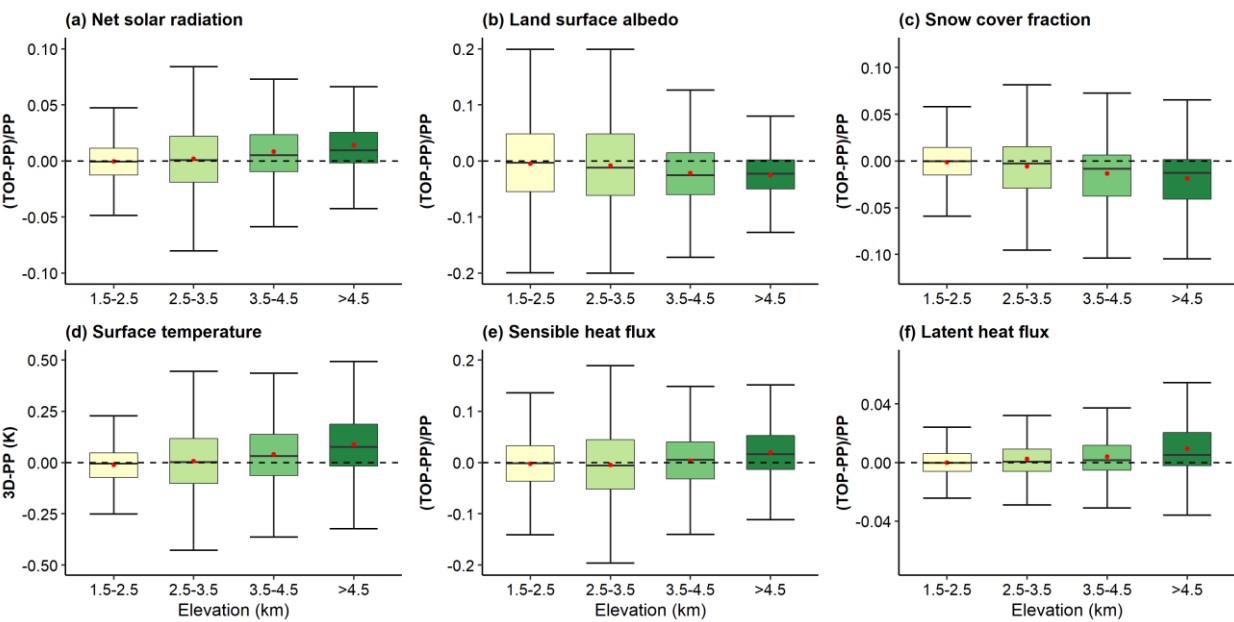

**Figure 15.** Boxplots of the relative (or absolute for surface temperature) differences of net solar radiation (a), land surface albedo (b), snow cover fraction (c), surface temperature (d), sensible heat flux (e) and latent heat flux (f) between TOP and PP in winter at four different elevations bands. Red points represent the mean values.

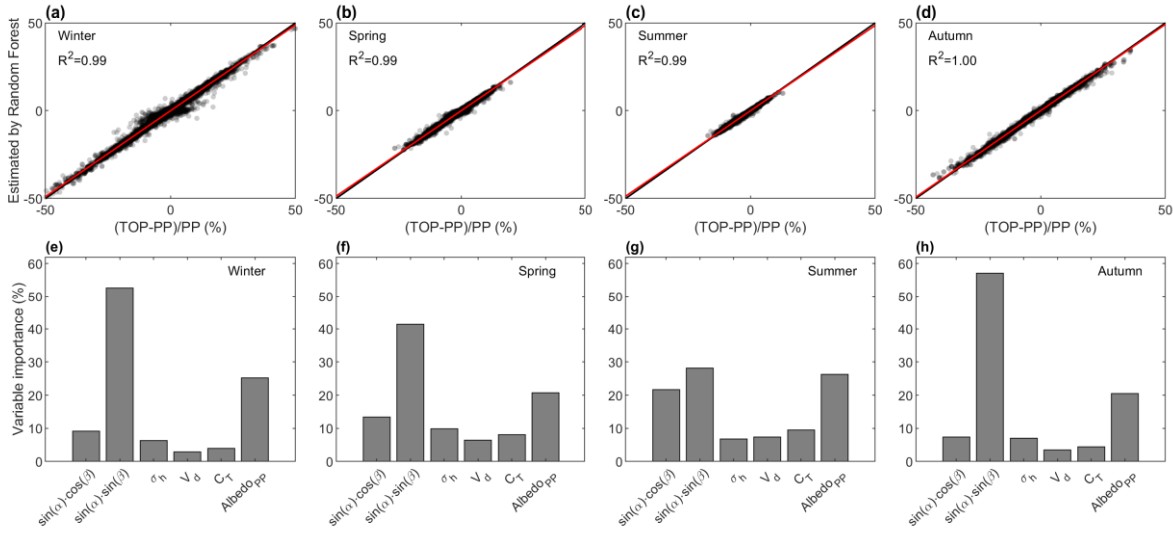

**Figure 16.** The performance of the random forest modeling in predicting the relative difference of land surface albedo between TOP and PP for four seasons (top row; a-d). The relative importance of topographic factors in predicting the differences in surface albedo between TOP and PP for four seasons (bottom row; e-h). $R^2$ is the coefficient of determination and the different topographic factors are described in the text.




### 3.3. Sensitivity to spatial scales

**The sub-grid topographic effects on surface energy balance, snow cover and surface temperature are sensitive to the spatial scales.** The sub-grid topographic effects on land surface albedo in winter show similar spatial patterns across spatial scales (Figure 17a-e). There are similar trends of the sub-grid topographic effects on land surface albedo with elevations at different spatial scales (Figure S7). Larger spatial heterogeneity in land surface albedo is present at finer spatial scales, but the pattern is smoothed at coarser spatial scales (Figures 17 and S7). As the spatial resolution becomes coarser, the terrain becomes flatter and thus the differences between TOP and PP are smaller. However, the relative difference between TOP and PP can still be large as -15% at coarse spatial scales (i.e., f19; Figure 17e). The statistical distributions of the relative differences in land surface albedo over the TP at different spatial scales are similar, with 0~-5% as the frequent value (Figure 17f). For snow cover, surface temperature and other energy balance variables, similar results are noted from Figures S2-S6 and the sub-grid topographic effects are still significant even at a spatial resolution as low as around 2°. For instance, for the spatial resolution of f19, the maximum values of the relative differences of net solar radiation, sensible heat and heat flux and snow cover fraction can be larger than 8%, 20%, 20% and -20%, respectively. The absolute difference of surface temperature for f19 is within 0.1K, but that for f09 is still large as 0.5K. In addition, the relative contributions of different topographic variables are similar at different spatial scales (Figure S8).

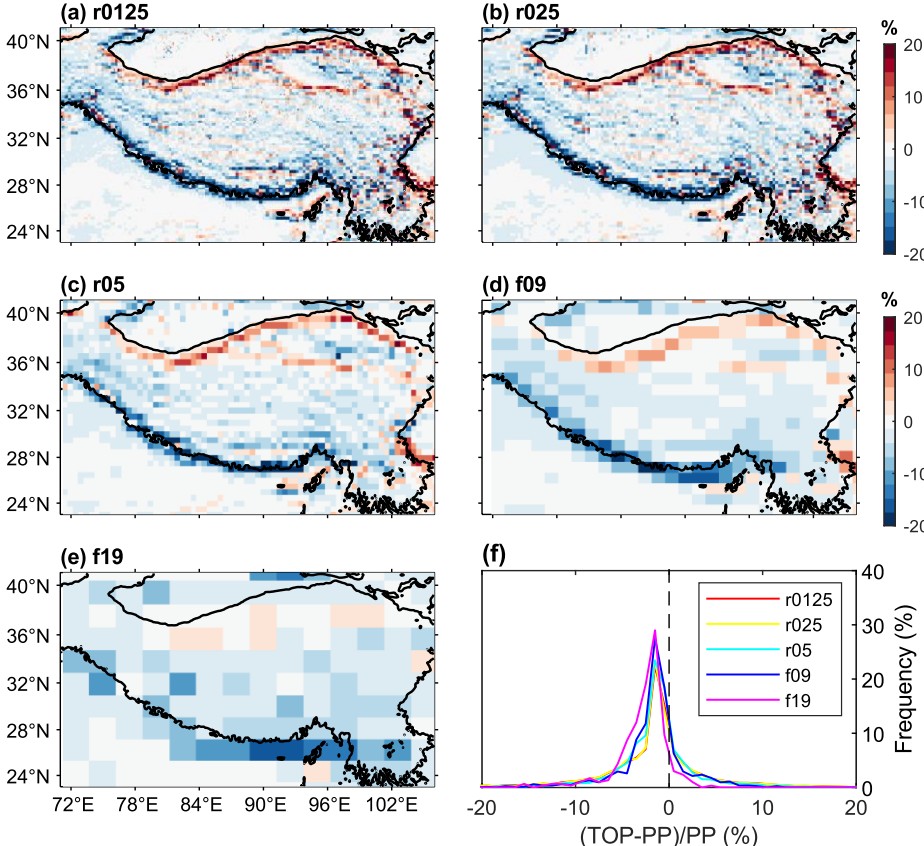

**Figure 17.** The relative differences of land surface albedo between TOP and PP in winter at different spatial scales (a-e) and the statistical histogram of their frequent distributions (f).





## 4. Discussion

Sub-grid topographic effects on solar radiation play an important role in surface energy balance, surface temperature and snowmelt over complex terrain. Simply neglecting the sub-grid topography can lead to large errors in simulating surface
energy balance. Compared to flat surfaces, the land surface albedo over the complex terrain of TP generally decreases and net solar radiation increases (Figures 9-10), which increases the surface temperature (Figure 12). Larger net solar radiation over mountainous regions increases sensible and latent heat fluxes (Figures 13-14) and decreases the snow cover areas due to increased snow melt and possibly snow-albedo feedback (Figure 11). The effects of sub-grid topography on solar radiation also show seasonal variations, which are more pronounced in winter, because **larger** solar zenith angles in winter over the
TP can cause stronger shadowing effects (Hao et al., 2018b) and large snow cover areas in winter can cause stronger reflected radiation from adjacent topography (Helbig et al., 2010). In addition, the sub-grid topographic effects are elevation-dependent (Figure 15), because mountain tops with higher elevations tend to receive more solar radiation due to the topographic effects **and thinner atmosphere**, while valley areas with lower elevations receive relatively less solar radiation due to the shadowing effects (Fan et al., 2019; Lee et al., 2015). Compared to PP, TOP produces results more consistent with
the MODIS observations, especially in the high-elevation and snow cover regions over the TP (Figures 2-8), which demonstrates that accounting for the sub-grid topographic effects over complex terrain improves the performance of ELM. In a high-resolution coupled model, the highly concentrated differences between TOP and PP along the southern edge of the TP could lead to important differences in simulating clouds, convection, terrain-induced circulation and transport of aerosols, with potentially important implications for modeling the South Asian monsoon and its hydrologic impacts. Future studies
including the sub-grid topographic effects in coupled simulations will address their impacts on coupled land-atmosphere processes.

Sub-grid topographic effects are strongly dependent on spatial scales. The sub-grid topographic effects are more pronounced at the finer resolution (Figures 17 and S2-S6) and tend to be spatially smoothed at a coarse resolution (Lee et al., 2011, 2013). Therefore, it is necessary to consider the sub-grid topographic effects on solar radiation in high-resolution land
surface modeling. However, the relative differences in net solar radiation between TOP and PP can still reach up to 8% in some regions even at the coarse spatial resolution of 2° (Figure S2). This demonstrates that the sub-grid topographic effects on solar radiation cannot be neglected even for simulations at coarse spatial resolutions.

Uncertainties of remote sensing data may affect their reliability as ground truth for evaluating the ELM simulations. The MODIS land surface albedo products have shown good consistencies with ground measurements (Moustafa et al., 2017;
Wang, 2004), but the algorithms used to derive the MODIS land surface albedo do not account for topography explicitly (Schaaf et al., 2002; Hao et al., 2020), which leads to large errors over rugged terrain (Hao et al., 2018a, 2018b). MODIS snow cover data has shown relatively poor performance when compared to ground measurements, especially over the regions of TP with higher elevation and shallower snow depth (Pu et al., 2007; Yang et al., 2015; Zhang et al., 2019). The accuracy of MODIS surface temperature products depends on the accuracy of land cover products and the prescribed surface
emissivity values (Duan et al., 2019). The MODIS evapotranspiration product is sensitive to the algorithm used to account for the environmental stresses over the TP, as well as, the atmospheric forcing data used to generate the product (Li et al., 2019b). The SRTM data, used to derive the topographic factors for the parameterization, has shown large errors in some regions (Grohmann, 2018; Mukherjee et al., 2013). More accurate topographic factors can be derived using globally consistent, high-quality DEM data such as the Copernicus 30-meter global Digital Elevation Model (GLO-30)
(https://spacedata.copernicus.eu). The quality of remote sensing data needs to be validated comprehensively before its use in evaluation of LSMs.

The inclusion of sub-grid topographic parameterizations in ELM improves the representations of surface energy balance to some degree, but many shortcomings in ELM's existing radiative transfer modeling scheme limit the potential for further improving the ELM simulations. The 1D two stream approximation method used in ELM represents the vegetation canopy
as a homogeneous "big leaf" (Yuan et al., 2017) and neglects the vertical multi-layer structure (Bonan et al., 2018) and the horizontal leaf clumping (Bailey et al., 2020; Braghiere et al., 2020; Li et al., 2019a). In the snow-covered regions, the ELM parameterizations for the effects of snow impurities (i.e., black carbon and dust mixing) on light scattering and absorption





processes need to be refined to account for internal mixing and non-spherical shapes of snow grains (Dang et al., 2019; He et al., 2018). In addition, ELM also does not account for the influence of adjacent terrain on longwave thermal radiation (Yan et al., 2020).


In this study, the same atmospheric forcings were used in the simulations at different spatial scales, which could be a source of error at a finer resolution (Fiddes and Gruber, 2014; Tesfa et al., 2020). Furthermore, the sub-grid parameterizations neglect the spatial correlation between sub-grid topography and plant functional types. The spatial pattern of vegetation types generally depends on the topographic distribution, which controls terrestrial water, energy, water, and carbon cycle (Reed et al., 2009). These aforementioned simplifications may affect the accurate representations of the sub-grid topographic effects on solar radiation in ELM at a coarse resolution. Combining the sub-grid topographic parameterizations implemented in ELM in the study with ELM's new sub-grid topography structure (Tesfa et al., 2017) and downscaling of atmospheric forcing (Tesfa et al., 2020) is anticipated to further improve the representations of the land surface processes at different spatial scales (Ke et al., 2013). A future study will investigate the impact of sub-grid topographic parameterization on the land-atmosphere interactions by performing ELM simulations with an active atmospheric model.



## 5. Conclusions

The computationally efficient sub-grid topographic parameterization on solar radiation of Lee et al. (2011) was implemented in ELM in this study. Results show that ELM simulations with the sub-grid topography parameterization have better agreements with the MODIS data for simulated surface energy balance, snow cover and surface temperature over the TP. Topography has significant effects on surface energy budget, snow cover as well as surface temperature that cannot be neglected. The absolute differences with and without accounting for sub-grid topography on net solar radiation, sensible heat flux, and latent heat flux exceed 20 W/m$^2$, 10 W/m$^2$, and 5 W/m$^2$, respectively. Similarly, the differences in land surface albedo, snow cover fraction, and surface temperature exceed 0.1, 20%, and 1K, respectively. Nearly all the relative differences of these variables, except surface temperature, reach up to 20%. The magnitude of the sub-grid topographic effects on solar radiation is seasonally-dependent and elevation-dependent, and is also sensitive to the spatial scales. Although the sub-grid topographic effects on solar radiation are more significant at finer spatial scales, they cannot be simply neglected even at coarse spatial scales. For example, the relative difference in land surface albedo when accounting for sub-grid topography in winter reaches up to -15% for the coarse spatial scale of 2$^0$. These results highlight the necessity of accounting for the sub-grid topographic effects in LSMs and show that our improvements in ELM are promising to advance understanding and modeling of the role of the surface topography on terrestrial processes.




**Code and data availability.** All remote sensing data are publicly accessible at the Google Earth Engine Platform (Gorelick et al., 2017). ELM codes are available publicly at https://github.com/E3SM-Project/E3SM (last access: 13 July 2020). Codes for sub-grid topographic improvements described in this paper is available at http://doi.org/10.5281/zenodo.4549401 (Hao, 2021) and codes to reproduce all results and plot all figures are publicly available at https://github.com/daleihao/Topographic_Effects.


**Author contributions.** DH designed the study, implemented the parameterization, performed the simulations, analyzed the results and drafted the original manuscript. GB designed the study, discussed the results and edited the manuscript. LRL edited the manuscript. All authors contributed to improving the manuscript.

**Competing interests.** The authors declare that they have no conflict of interest.


**Acknowledgements.** This research used resources of the National Energy Research Scientific Computing Center (NERSC), a DOE Office of Science User Facility supported by the Office of Science of the U.S. Department of Energy under Contract



No. DE-AC02-05CH11231. This research also used DOE's Biological and Environmental Research Earth System Modeling program's Compy computing cluster located at Pacific Northwest National Laboratory.

**Financial support.** This research has been supported by the U.S. Department of Energy Office of Science Biological and Environmental Research as part of the Earth and Environmental System Modeling program and the U.S. National Oceanic and Atmospheric Administration (NOAA) under award numbers NOAA-OAR-CPO-2019-2005530 and NA19OAR4310243. The reported research was conducted at Pacific Northwest National Laboratory, which is operated for the U.S. Department of Energy by Battelle Memorial Institute under contract DE-AC05-76RL01830.

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
