# Peer review of "A Parameterization of Sub-grid Topographical Effects on Solar Radiation in the E3SM Land Model (Version 1.0): Implementation and Evaluation Over the Tibetan Plateau"

_Geoscientific Model Development, 2021_

## Referee Comment (RC1)

**General comments**

This manuscript uses an existing parameterization for sub-grid topographical effects on solar surface irradiance to show how accounting for sub-grid topography, instead of assuming a flat surface, affects the albedo, energy balance and temperature of the surface. The authors find that accounting for sub-grid topography can have large effects on the land surface model, depending on season, elevation and grid resolution. Furthermore, the authors use MODIS data to evaluate this newly implemented parameterization and find that accounting for sub-grid scale topography generally improves the representation of the surface.

The paper is interesting within the scope of the journal. However, I think the presentation of the manuscript can be improved. First of all, I would suggest to switch the order of sections 3.1 and 3.2, e.g. start with the effects of sub-grid topography on the land surface and then show to what extent this improves the results with respect to MODIS. Furthermore, I suggest to strongly reduce the number of figures in the manuscript. Figures 2-7 and 9-14 fill up a lot of space and are a bit repetitive. It certainly helps the reader to keep a few maps, but I would suggest to use a more concise visualization (e.g. such as in Figure 8) of the data.

I recommended major revisions based on the main comments above and the specific comments and technical corrections listed below.

**Specific comments**

1. p. 2, ln 66-68: Mention clearly where your evaluation and analysis of the sub-grid scale topography effects differs from Lee et al. (2019), besides the use of a different land surface model.
2. p. 3, ln 85-87: State more clearly that you use this parametrization to investigate for different seasons how the large the effects of using a sub-grid scale topography parametrization on solar irradiance are. And that both the simulation with and without this parametrization are compared to MODIS to investigate to what extent using this parametrization improves the land surface.
3. p. 4, ln. 99. "multi-scattered radiation": what is this term exactly, isn't it also part of the diffuse irradiance?
4. p. 5, ln. 151: "TP, known as the Third Pole" may suggest TP is the abbreviation of Third Pole, which is confusing. Suggestion: "The Tibetan Plateau (TP), also known as the Third Pole,"
5. p. 5, ln 161: I suppose that with "offline" you mean that ELM is not coupled to the atmospheric/oceanic model? If so, how do you derive the radiative fluxes at the top of the canopy, i.e. what is the atmospheric input for these RT calculations and are these similar for TOP and PP?
6. p. 6, ln. 188: What is the reason for these two times (10:30, 22:30)?
7. p. 6, ln. 181: "Relative difference" suggests something like (TOP-MODIS)/MODIS and (PP-MODIS)/MODIS, whereas the computed quantity seems to be the change in the bias with respect to MODIS between TOP and PP.
8. p. 6, ln. 193-195: How is the evolution of the snow cover determined in ELM? Besides snow cover, wouldn't snow depth be interesting as well?
9. p. 6, ln. 196: How valid is that assumption, given that you have seasonal changes in vegetation and snow cover as well as snow cover differences between the simulation.?
10. p. 7, ln 205: Is all data used to construct the random forests?
11. p. 7, ln 220-221: Can you explain why diffuse albedo is worse with sub-grid topographic effects?
12. p. 7, ln. 227 & ln 232 (and a few other locations in the manuscript): What does significant mean here?
13. p. 7, ln. 231: Looking at figure 6, the difference between TOP and PP in some regions actually does not seem to be very small in winter and spring.

14. p. 8. ln 242: Does significant mean statistically significant or that the bias difference is small compared to the mean surface temperature? Also, the fact that the nighttime surface temperature in the upper two elevation bands performs worse with TOP for most of the region is interesting and worth mentioning. Do you have an explanation why it may be worse?

15. p. 10, ln. 272-273: ", while the" You compare a difference in W/m2 to a relative difference (%), which can be confusing.

16. p. 10, ln. 279-280: Can you elaborate on this? Does it mean that the standard deviation of elevation can explain most of the sub-grid scale topography effects on the albedo?

17. p. 10. ln. 282-283: "the absolute .. and -20%" The units and signs are a bit confusing here. First of all, I suggest to express snow cover fraction as a number between 0 and 1 instead of a percentage, to make the distinction between absolute and relative difference more clear. Also, "larger than" can be confusing because the difference is negative. You could rephrase as "the decrease in .... can be larger than ..."

18. p. 11, ln. 296: "in winter", how are the elevation-dependent patterns in the other seasons compared to the winter?

19. p. 11, ln. 304: Shouldn't "relative differences" be "absolute differences", given the ±0.1? Also, it is not really clear to me what the exception ( "except") described in this part of the sentences exactly is.

20. p. 11, ln. 308: "sensitivity analysis" can you elaborate on this (e.g what kind of analysis)?

21. p. 16, ln. 365: since the relative difference in snow cover fraction is negative (-20%), "maximum value" can be somewhat confusing for this quantity.

22. P. 16, Figure 17 (and Figures S2-S6): the location of the red line (r0125) is important because it is serves as the 'reference', but the line is sometimes hard to see.

23. p. 17, ln.398-408 & sections 2.4/2.5: Are you able to compare typical errors estimates of MODIS with the differences between TOP and PP?

**Technical corrections**

24. p. 7, ln 205: calculations -> calculation

25. p. 7, ln 221; p. 10, ln 284; p. 10, ln 285 and more: "difference(s) of" -> "difference(s) in"

26. p. 7, ln 222: : "regions"->"region", or perhaps: "the whole rectangular regions" -> "the whole domain"

27. p. 10, ln. 277 & ln. 289: "different seasons" -> "all seasons"?

28. p. 10, ln.291: this 20% has already been mentioned in line 289.

29. p. 11, ln. 297: "but" -> "therefore"?

30. p. 11, ln. 300: "With the increase of elevation bands" -> "At higher elevations"

31. p. 11, ln. 301: "to larger" -> "to a larger"

32. p. 16, ln. 357: Figure S7 is referenced before figures S2-S6.

33. p. 17, ln. 379 & ln. 383: Why are "larger" and "and thinner atmosphere" in bold?

34. p. 18, ln 427: "the study" -> "this study"

35. p. 18, ln 432: "on solar radiation" -> "for solar radiation"

---

## Author Comment (AC1)

**Reviewer 1**

**General comments**

This manuscript uses an existing parameterization for sub-grid topographical effects on solar surface irradiance to show how accounting for sub-grid topography, instead of assuming a flat surface, affects the albedo, energy balance and temperature of the surface. The authors find that accounting for sub-grid topography can have large effects on the land surface model, depending on season, elevation and grid resolution. Furthermore, the authors use MODIS data to evaluate this newly implemented parameterization and find that accounting for sub-grid scale topography generally improves the representation of the surface.

The paper is interesting within the scope of the journal. However, I think the presentation of the manuscript can be improved. First of all, I would suggest to switch the order of sections 3.1 and 3.2, e.g. start with the effects of sub-grid topography on the land surface and then show to what extent this improves the results with respect to MODIS. Furthermore, I suggest to strongly reduce the number of figures in the manuscript. Figures 2-7 and 9-14 fill up a lot of space and are a bit repetitive. It certainly helps the reader to keep a few maps, but I would suggest to use a more concise visualization (e.g. such as in Figure 8) of the data.

I recommended major revisions based on the main comments above and the specific comments and technical corrections listed below.

Thank you very much for your useful suggestions/comments. As suggested, we reorganized the order of the sections and in the revised manuscript, the results sections include **3.1 Sub-grid topographic effects on surface energy budget, surface temperature and snow cover/depth, 3.2 Contributions of different factors, 3.3 Sensitivity to elevations, 3.4 Sensitivity to spatial scales, and 3.5 Comparison with MODIS data**. We also reduced the number of figures by using more concise figures to show the results clearly in the revised manuscript. A detailed response to your specific comments is provided below.

**Specific comments**

**1. p. 2, ln 66-68: Mention clearly where your evaluation and analysis of the sub-grid scale topography effects differs from Lee et al. (2019), besides the use of a different land surface model.**

Lee et al. (2019) focused on winter at a horizontal spatial resolution of 0.9° × 1.25°, while our study analyzed the topographic effects at various spatial scales from high resolution (0.15°) to coarser resolution (about 2°), for all four seasons. Lee et al. (2019) compared the surface downward solar flux, surface upward solar flux, surface net solar flux, surface albedo, and air temperature of the model simulation with those of the CERES observations and CMIP5 models, while our study focused on land surface processes and comparison of high-resolution MODIS data and model simulations for direct albedo, diffuse albedo, snow cover, surface daytime/nighttime temperature, and latent heat flux. In addition, we also analyzed the sensitivity to elevation and the contributions of different factors. We clarified these in Line 69-72 of the revised manuscript.

**2. p. 3, ln 85-87: State more clearly that you use this parametrization to investigate for different seasons how the large the effects of using a sub-grid scale topography parametrization on solar irradiance are. And that both the simulation with and without this parametrization are compared to MODIS to investigate to what extent using this parametrization improves the land surface.**

We stated these more clearly as suggested in Line 89-94 of the revised manuscript as below: The sub-grid topographic effects on surface energy balance, snow cover/depth and surface temperature were investigated based on the ELM simulations. The contribution of different factors to the sub-grid topographic effects and the dependence of the sub-grid topographic effects on seasons, elevations and spatial scales were also analyzed. A suite of remotely sensed data from the Moderate Resolution Imaging Spectroradiometer (MODIS) were used to compare with the ELM simulations with different parameterizations for solar radiation in different seasons.

**3. p. 4, ln. 99. "multi-scattered radiation": what is this term exactly, isn't it also part of the diffuse irradiance?**

We replaced it with 'coupled radiation that represents surface reflected radiation that is further reflected or scattered by atmospheric particles', in Line 107-108 of the revised manuscript. The incoming solar radiation for a flat surface is composed of direct radiation ($F_{dir}^{PP}$) from sun, diffuse radiation ($F_{dif}^{PP}$) from sky, and coupled radiation ($F_{couple}^{PP}$). ELM-v1.0 assumes flat surfaces and accounts for $F_{dir}^{PP}$ and $F_{dif}^{PP}$, while neglecting $F_{couple}^{PP}$.

**4. p. 5, ln. 151: "TP, known as the Third Pole" may suggest TP is the abbreviation of Third Pole, which is confusing. Suggestion: "The Tibetan Plateau (TP), also known as the Third Pole,"**

We revised the manuscript as suggested in Line 170.

**5. p. 5, ln 161: I suppose that with "offline" you mean that ELM is not coupled to the atmospheric/oceanic model? If so, how do you derive the radiative fluxes at the top of the canopy, i.e. what is the atmospheric input for these RT calculations and are these similar for TOP and PP?**

Yes, we only ran the ELM model and will do the coupling run in the future study. The 3-hourly Global Soil Wetness Project meteorological forcing data set version 1 (GSWP3v1) (Dirmeyer et al., 2006; Yoshimura and Kanamitsu, 2013) with 0.5°×0.5° spatial resolution was used to drive all the model simulations. The bilinear interpolation techniques were used to downscale the GSWP3v1 data into the required spatial resolution, and the coszen (i.e., the cosine of the solar zenith angle)-based, nearest neighbor, and linear interpolation methods were used to downscale the solar, precipitation and other data to the half-hourly temporal resolution, respectively. We clarified these in Line 180-190 of the revised manuscript.

**6. p. 6, ln. 188: What is the reason for these two times (10:30, 22:30)?**

The MODIS sensor onboard the Terra satellite has an overpass time of 10:30 am (local solar time) and 10:30 pm (local solar time), respectively. Thus, we selected those two times for model evaluation. We have revised the manuscript accordingly in Line 235 to include

the MODIS instantaneous surface temperature data was derived for daytime and nighttime corresponding to the MODIS overpass time: 10:30 and 22:30 (local solar time), respectively.

**7. p. 6, ln. 181: "Relative difference" suggests something like (TOP-MODIS)/MODIS and (PPMODIS)/MODIS, whereas the computed quantity seems to be the change in the bias with respect to MODIS between TOP and PP.**

We revised the manuscript as suggested in Line 238.

**8. p. 6, ln. 193-195: How is the evolution of the snow cover determined in ELM? Besides snow cover, wouldn't snow depth be interesting as well?**

As suggested, we added the related introduction about the snow processes in ELM in Line 97-101 as below:

ELM (Version 1.0) is based on the Community Land Model Version 4.5 (CLM4.5) (Golaz et al., 2019). ELM calculates canopy radiation flux using the two-stream approximation methods, snow albedo using the Snow, Ice, and Aerosol Radiative Model (SNICAR) model (Flanner et al., 2007), and snow cover fraction based on snow water equivalent (Swenson and Lawrence, 2012). ELM also represents the snow hydrological processes including snowfall accumulation, melting, refreezing, compaction, aging, water transfer across layers, etc.

We also added the comparison of snow depth in Line 255-256 of the revised manuscript. In ELM, snow depth has a positive correlation to snow cover fraction which is calculated based on snow water equivalent (Bonan et al., 2019). The results show that larger net solar radiation over mountainous regions increases sensible and latent heat fluxes and decreases the snow cover fractions and snow depth due to increased snow melt and possibly snow-albedo feedback (Figure

4), which may alleviate the snow depth overestimation over the TP in ESMs (Wei and Dong, 2015).

Wei, Z. and Dong, W.: Assessment of Simulations of Snow Depth in the Qinghai-Tibetan Plateau Using CMIP5 Multi-Models, Arctic, Antarctic, and Alpine Research, 47, 611-625, 2015.

Bonan, G., 2019. Climate change and terrestrial ecosystem modeling. Cambridge University Press.

**9. p. 6, ln. 196: How valid is that assumption, given that you have seasonal changes in vegetation and snow cover as well as snow cover differences between the simulation.?**

In ELM, snow emissivity is fixed as 0.97, soil emissivity is fixed as 0.96, and vegetation emissivity is around 0.98 which depends on the specific leaf/stem area index. For a grid with different land cover types, the assumption that the grid emissivity is 1.0 is reasonable and is also used in the land-atmosphere interaction of E3SM. In addition, sub-grid variability of emissivity is not considered in the sub-grid topographic parameterization for solar radiation and we only aim to compare the differences between TOP and PP rather than the absolute accuracy of the calculated surface temperature.

**10. p. 7, ln 205: Is all data used to construct the random forests?**

Yes. Combined with the driving variables, all the ELM-derived seasonally-averaged data was used to train the random forest model and measure the relative importance of different factors in controlling the sub-grid topographic effects. We clarified these in Line 211-212 of the revised manuscript.

**11. p. 7, ln 220-221: Can you explain why diffuse albedo is worse with sub-grid topographic effects?**

We provided some possible reasons to explain the inconsistencies between ELM simulations and MODIS data, especially for diffuse albedo and nighttime surface temperature in Line 422-454 of the revised manuscript as below:

1) **For the ELM model:** The inclusion of sub-grid topographic parameterizations for solar radiation in ELM improves the representations of surface energy balance to some degree, but many shortcomings in ELM's existing radiative transfer modeling scheme limit the potential for further improving the ELM simulations. The 1D two stream approximation method used in ELM represents the vegetation canopy as a homogeneous "big leaf" (Yuan et al., 2017) and neglects the vertical multi-layer structure (Bonan et al., 2018) and the horizontal leaf clumping (Bailey et al., 2020; Braghiere et al., 2020; Li et al., 2019a). In the snow-covered regions, the ELM parameterizations for the effects of snow impurities (i.e., black carbon and dust mixing) on light scattering and absorption processes need to be refined to account for internal mixing and non-spherical shapes of snow grains (Dang et al., 2019; He et al., 2018).

2) **For the remote sensing data:** MODIS data also has some uncertainties related to the retrieval algorithm over rugged terrain, sensor calibration, atmospheric correction, etc. Uncertainties of remote sensing data may affect their reliability as ground truth for evaluating the ELM simulations. The MODIS land surface albedo products have shown good consistencies with ground measurements (Moustafa et al., 2017; Wang, 2004), but the semi-empirical kernel-driven-model-based algorithms used to derive the MODIS land surface albedo do not account for topography explicitly (Schaaf et al., 2002; Hao et al., 2020), which leads to large errors over rugged terrain (Hao et al., 2018a, 2018b). MODIS snow cover data has shown relatively poor performance when compared to ground measurements, especially over the regions of TP with higher elevation and shallower snow depth (Pu et al., 2007; Yang et al., 2015; Zhang et al., 2019). The accuracy of MODIS surface temperature products depends on the accuracy of land cover products and the prescribed surface emissivity values (Duan et al., 2019). The MODIS evapotranspiration product is sensitive to the algorithm used to account for the environmental stresses over the TP, as well as, the atmospheric forcing data used to generate the product (Li et al., 2019b).

**12. p. 7, ln. 227 & ln 232 (and a few other locations in the manuscript): What does significant mean here?**

We replaced this somewhat subjective word as 'small' or 'large' throughout the revised manuscript to make it clearer.

**13. p. 7, ln. 231: Looking at figure 6, the difference between TOP and PP in some regions actually does not seem to be very small in winter and spring.**

We rephrased this sentence as "the difference between TOP and PP is small in summer and autumn but large in winter and spring" in Line 364-365. In addition, we also give the possible explanation in the discussion part (Please also see our responses to Questions #11).

**14. p. 8. ln 242: Does significant mean statistically significant or that the bias difference is small compared to the mean surface temperature? Also, the fact that the nighttime surface temperature in the upper two elevation bands performs worse with TOP for most of the region is interesting and worth mentioning. Do you have an explanation why it may be worse?**

We replaced "significant" as "small" or "large" throughout the revised manuscript to make it clearer.

We mentioned these results in Line 378 as: "For nighttime surface temperature, the difference in biases increases with elevation". In addition, the related processes in ELM still have some uncertainties (see our responses to Question #11). These may partly explain the inconsistencies between ELM simulations and MODIS data, especially for diffuse albedo and nighttime surface temperature (Figure 10).

**15. p. 10, ln. 272-273: ", while the" You compare a difference in W/m2 to a relative difference (%), which can be confusing.**

We revised the manuscript for clarity in Line 244.

**16. p. 10, ln. 279-280: Can you elaborate on this? Does it mean that the standard deviation of elevation can explain most of the sub-grid scale topography effects on the albedo?**

Here we want to show that the sub-grid topographic heterogeneity is related to the spatial patterns of the difference between TOP and PP. We revised this in Line 250-252 as "the spatial pattern of the difference in land surface albedo between TOP and PP is similar to the heterogeneous spatial pattern of topography (Figure 2)".

**17. p. 10. ln. 282-283: "the absolute .. and -20%" The units and signs are a bit confusing here. First of all, I suggest to express snow cover fraction as a number between 0 and 1 instead of a percentage, to make the distinction between absolute and relative difference more clear. Also, "larger than" can be confusing because the difference is negative. You could rephrase as "the decrease in .... can be larger than ..."**

As suggested, we expressed the snow cover fraction as a number between 0 and 1 instead of a percentage, and revised the manuscript to avoid the confusion in Line 255.

**18. p. 11, ln. 296: "in winter", how are the elevation-dependent patterns in the other seasons compared to the winter?**

As suggested, we analyzed the elevation-dependent patterns in four seasons, taking the land surface albedo as an example and included an additional figure in the manuscript (Figure 6). The results show that these elevation-dependent patterns are similar for all seasons, although the differences between TOP and PP are larger in winter than in summer.

**19. p. 11, ln. 304: Shouldn't "relative differences" be "absolute differences", given the ±0.1? Also, it is not really clear to me what the exception ( "except") described in this part of the sentences exactly is.**

We revised "0.1" as "10%" to represent the relative difference, and replaced "except that" with "and" to avoid the confusion in Line 315-316.

**20. p. 11, ln. 308: "sensitivity analysis" can you elaborate on this (e.g what kind of analysis)?**

We replaced "sensitivity analysis" with "variable importance analysis" in Line 286 to clarify that here we aimed to analyze the contributions of different factors to the differences between TOP and PP.

**21. p. 16, ln. 365: since the relative difference in snow cover fraction is negative (-20%), "maximum value" can be somewhat confusing for this quantity.**

We deleted this expression in the revised manuscript. Please see the responses to Question #17 for our revisions.

**22. P. 16, Figure 17 (and Figures S2-S6): the location of the red line (r0125) is important because it is serves as the 'reference', but the line is sometimes hard to see.**

We have adjusted the order of different lines in these figures in the revised manuscript, to make the 'reference' line to be seen easily.

**23. p. 17, ln.398-408 & sections 2.4/2.5: Are you able to compare typical errors estimates of MODIS with the differences between TOP and PP?**

As suggested, we have compared the typical errors of MODIS data with the differences between TOP and PP in Line 431-440 of the revised manuscript, as below:

However, the topography-induced differences between TOP and PP can be comparable to the errors of MODIS data. For example, Wang et al. (2004) reported that compared to ground

measurements, MODIS albedo had a maximum error of 0.036 in a semidesert region on the TP, which is smaller than the maximum difference of 0.1 between TOP and PP (Figure 4). Wang et al. (2007) showed that the mean and maximum errors of MODIS surface temperature were 0.27 K and 2.61 K, respectively at a semi-desert site on the western TP, which is comparable to the maximum difference of 1 K between TOP and PP (Figure 4). Salomonson and Appel (2004) showed that using the Landsat 30 m observations as the benchmark, the mean error of MODIS snow cover fraction was smaller than 0.1, which is comparable to the difference of 0.1 between TOP and PP (Figure 4). Mu et al. (2007) showed that the 8-day MODIS latent heat flux had a mean bias from -5.8 to 39.9 $W/m^2$, possibly larger than the difference between TOP and PP in our study (Figure 4).

**Technical corrections**

24. p. 7, ln 205: calculations -> calculation

25. p. 7, ln 221; p. 10, ln 284; p. 10, ln 285 and more: "difference(s) of" -> "difference(s) in"

26. p. 7, ln 222: : "regions"->"region", or perhaps: "the whole rectangular regions" -> "the whole domain"

27. p. 10, ln. 277 & ln. 289: "different seasons" -> "all seasons"?

28. p. 10, ln.291: this 20% has already been mentioned in line 289.

29. p. 11, ln. 297: "but" -> "therefore"?

30. p. 11, ln. 300: "With the increase of elevation bands" -> "At higher elevations"

31. p. 11, ln. 301: "to larger" -> "to a larger"

32. p. 16, ln. 357: Figure S7 is referenced before figures S2-S6.

33. p. 17, ln. 379 & ln. 383: Why are "larger" and "and thinner atmosphere" in bold?

34. p. 18, ln 427: "the study" -> "this study"

35. p. 18, ln 432: "on solar radiation" -> "for solar radiation"

We have revised the above sentences carefully as suggested and checked/revised the grammar/typo issues throughout the revised manuscript.

---

## Author Comment (AC2)

**Reviewer 2**

The authors incorporated a sub-grid topographic parameterization in the E3SM Land Model (ELM) to quantify the effects of sub-grid topography on solar radiation flux, which includes the shadow effects and multi-scattering between adjacent terrain. They found that incorporating the sub-grid topographic effects generally reduces the biases of ELM in simulating surface energy balance, snow cover and surface temperature particularly in the high-elevation and snow-cover regions over the TP. Overall, this manuscript is well organized and written. However, there are still a few places that require further clarifications and discussions. Please see my specific comments below.

Thank you very much for these useful suggestions/comments. We have revised the manuscript carefully.

**Specific comments**
**1. I suggest being more specific and accurate about "sub-grid topographic parameterizations". This study actually focused on the subgrid terrain-radiation interactions instead of other subgrid topographic effects.**

As suggested, we have revised "sub-grid topographic parameterizations" as "sub-grid topographic parameterizations for solar radiation" throughout the revised manuscript, to more accurately match our study.

**2. The authors mentioned that ELM uses a novel topography-based sub-grid spatial structure. How does this new sub-grid spatial structure interact with the implemented subgrid radiation parameterization? Are they coupled?**

The spatial pattern of vegetation types generally depends on the topographic distribution, which controls terrestrial water, energy, water, and carbon cycle (Reed et al., 2009). These

aforementioned simplifications may affect the accurate representations of the sub-grid topographic effects on solar radiation in ELM at a coarse resolution. Combining the sub-grid topographic parameterizations for solar radiation implemented in ELM in this study with ELM's new sub-grid topography structure (Tesfa et al., 2017) and downscaling of atmospheric forcing (Tesfa et al., 2020) is anticipated to further improve the representations of the land surface processes at different spatial scales (Ke et al., 2013). We stated these in Line 457-463 of the revised manuscript. We will further couple them in the future study, but this is out of the scope of this manuscript.

**3. I suggest providing a schematic figure showing different flux components (Section 2.2) for the parameterization.**

As suggested, we have added a schematic diagram for different flux components as Figure 1 in the revised manuscript.

**4. Section 2.2: the original parameterization includes a coupled flux term, which however was not included in the implementation (e.g., Eqs 10-11). Any specific reasons? How much impact would this missing of the coupled term have on simulation results?**

We did not include the coupled component in the current parameterization because:

1) The impact of the coupled component on the total radiation is not significant in many cases because the magnitude of the variation in the deviation of the coupled flux is only about 0.5 $W/m^2$, while the deviations in the total surface solar flux are on the order of 100 $W/m^2$ (Lee et al., 2011).

2) The coupled flux is more complicated because it contains photons experiencing multiple scattering and reflection, and is not linearly proportional to surface albedo. We found that the regression analysis was less satisfactory in cases of low albedo values and we plan to include the coupled term using higher resolution data in future work.

We clarified these in Line 164-165 of the revised manuscript.

**5. The implementation adjusts albedo to account for the subgrid radiation effect. What is the rationale and justification to make this assumption? In theory, the surface albedo is a land surface intrinsic property, and by accounting for the additional subgrid terrainradiation fluxes (e.g., reflected from neighboring terrain), the change should be in the incoming solar radiation instead of surface albedo.**

For the offline simulations of the land model, the adjustments of incoming solar radiation and land surface albedo are identical theoretically, which motivated the idea to adjust albedo instead of downward radiation. However, for coupled simulations of the land and atmospheric models, only adjusting the surface downward/upward flux will lead to inconsistencies between the surface and the first level of the atmosphere above the surface. This is because the atmosphere model takes land surface albedo as the lower boundary condition rather than the upwelling solar radiation flux computed by the land model. Thus, we need to modify land surface albedo for consistency in fully coupled simulations.

Specifically, in the structure of a global climate model, the land surface model computes the surface albedo, taking into account land types, snow cover, soil moisture, and other factors. This albedo is then employed as a boundary condition in the global climate model for radiative transfer calculations. We can use the sub-grid topographic parameterization for solar radiation to adjust the land surface albedo, i.e. the ratio of the upward flux to the downward flux, such that the downward flux adjustment remains unchanged. In this manner, a balance of the total energy flux at the surface would be ensured, which is critical for long-term climate simulations (Lee et al., 2015). We clarified these in Line 153-155 of the revised manuscript.

In addition, the adjusted land surface albedo in our methods is closer to the apparent land surface albedo, observed by satellite remote sensing.

Lee, W.-L., Gu, Y., Liou, K. N., Leung, L. R. and Hsu, H.-H.: A global model simulation for 3-D radiative transfer impact on surface hydrology over Sierra Nevada and Rocky Mountains, Atmospheric Chemistry and Physics Discussions, 14(22), 31603–31625, doi:10.5194/acpd-14-31603-2014, 2015.

**6. Do the fitting parameters (A) in the subgrid radiation parameterizations vary across different scales? What are the values for the fitting parameters? A table listing these values would be good. What is the applicable range of spatial scales for the subgrid parameterization?**

As suggested, we have added Tables S1-S2 to list the values of the fitted parameters. In the current parameterization, we used the Shuttle Radar Topography Mission (SRTM) global data set at a resolution of 90 m to perform 3-D Monte Carlo photon tracing simulations to improve parameterization accuracy. The parameterization was developed at a 10 km × 10 km spatial scale, representative of a grid size of traditional weather models. Lee et al. (2013) demonstrated that the parameterization can be applied to various spatial resolutions larger than 10 km × 10 km. We added these descriptions in Line 164-168 of the revised manuscript.

**7. Some clarifications and descriptions are needed in Section 2.3. (1) What satellite data is used for LAI? (2) What are the native spatial and temporal resolutions of GSWP3v1 data and how did the authors interpolate the data to different simulation resolutions? (3) Since the authors focused on the analysis on snow and related surface quantities, a description of how ELM handles key snow processes and properties needs to be included.**

As suggested, we clarified the above in Line 97-104 of Sections 2.1 and Line 180-190 of 2.3 in the revised manuscript, as below:

1) In Section 2.3, MODIS LAI data was used.
2) The GSWP3v1 data has a spatial resolution of 0.5 degrees and temporal resolution of 3-hourly. The bilinear interpolation technique was used to downscale the GSWP3v1 data to the required spatial resolution, and the coszen (i.e., the cosine of the solar zenith angle)-based, nearest neighbor, and linear interpolation methods were used to downscale the solar, precipitation and other data to the half-hourly temporal resolution, respectively.

3) For the related processes, ELM (Version 1.0) is based on the Community Land Model Version 4.5 (CLM4.5) (Golaz et al., 2019). ELM calculates canopy radiation flux using the two-stream approximation methods, snow albedo using the Snow, Ice, and Aerosol Radiative Model (SNICAR) model (Flanner et al., 2007), and snow cover fraction based on snow water equivalent (Swenson and Lawrence, 2012). ELM also represents snow hydrological processes including accumulation, melt, compaction, aging, and water transfer across layers.

**8. The authors used a random forest model to quantify the sensitivity of topographic factors. Why not directly use the physics-based ELM model and vary those topographic factors to do the sensitivity tests? To me, the random forest model itself introduces additional uncertainties in the analysis.**

ELM is computationally too expensive to be directly use for performing the global sensitivity analysis covering different atmospheric conditions, soil and vegetation characteristics. Besides, simply varying selected variables in a fixed range may lead to unrealistic combinations of variables deviating from the real world. Therefore, we combined the ELM simulations over the Tibetan Plateau and a random forest model to evaluate the variable importance.

We also further clarified the theory of random forest models used to measure the variable importance in Line 205-209 of Section 2.4 in the revised manuscript as: The random forest model is a regression tree-based bootstrapped non-parametric machine learning model, which allows the calculation of the variable importance by estimating the out-of-bag (OOB) errors (Breiman, 2001). The OOB error represents the mean prediction error for each sample $x_i$, which uses only the trees that did not have $x_i$ in their bootstrap sample. To measure the importance of the j-th feature for training, the values of the j-th feature are permuted among the training data and the OOB error is computed for each perturbed data set. The importance score for the j-th feature is computed by averaging the difference in the OOB error before and after the permutation over all trees.

In addition, we tested the correlation between different factors and the sub-grid topographic effects based on the ELM simulations. Different factors show different correlations (both sign and magnitude) with the relative difference in land surface albedo between TOP and PP, as shown in the figure below. These demonstrate that different factors have different contributions which vary with seasons and are consistent with the variable-importance analysis based on the random forest model.

[Figure]

Figure. Correlation coefficients (Rs) between different factors and the relative difference in land surface albedo between TOP and PP for four seasons.

**9. I am a little concerned about the evaluation of surface albedo using MODIS albedo data. (1) Note that MODIS data is retrieved through algorithms that only assume planeparallel radiative transfer. So it may not be reasonable to use MODIS albedo as a justification for the subgrid terrain-radiation improvement. (2) Also, it is not clear how much improvement in surface albedo comes from the direct treatment of subgrid radiation and how much comes from the snow cover improvement.**

   **For question (1):**
   Indeed, there are uncertainties in the MODIS products especially in rugged terrain and we have now expanded Section 4 to include a discussion regarding those uncertainties in Line 422-444. Therefore, we did not use MODIS data as a benchmark and only aimed to compare the ELM simulations with MODIS data to reveal the sub-grid topographic effects in the revised manuscript.

For instance, the operational MODIS albedo algorithms use a semi-empirical kernel-driven model (Schaaf et al., 2012):

$$R(\Omega, \lambda) = f_{iso}(\lambda) \cdot K_{iso}(\Omega) + f_{vol}(\lambda) \cdot K_{vol}(\Omega) + f_{geo}(\lambda) \cdot K_{geo}(\Omega) \qquad (1)$$

where $f_{iso}$, $f_{vol}$, and $f_{geo}$ are three empirical kernel parameters, and $K_{iso}$, $K_{vol}$, and $K_{geo}$ are isotropic, volumetric-scattering, and surface scattering kernels, respectively. Generally, $K_{iso}$ is set to 1, and $K_{vol}$ and $K_{geo}$ are derived from complex radiative transfer and geometric optical models. These radiative transfer and geometric optical models used the plane parallel assumptions. Specifically, the algorithms first calculate the three kernel parameters by fitting the multi-angular reflectance, and then calculate the albedo by the hemispherical integration based on equation (1). Although the kernels $K_{iso}$, $K_{vol}$, and $K_{geo}$ don't account for topographic effects, the fitted kernel parameters can be affected by topography because the topography has large effects on the observed reflectance. Therefore, the MODIS algorithms do not account for topography explicitly. We clarified these aforementioned points in the discussion of the revised manuscript.

We also compared the typical errors of MODIS data with the differences between TOP and PP in Line 431-440 of the revised manuscript, as below:

However, the topography-induced differences between TOP and PP can be comparable to the errors of MODIS data. For example, Wang et al. (2004) reported that compared to ground measurements, MODIS albedo had a maximum error of 0.036 in a semidesert region on the TP, which is smaller than the maximum difference of 0.1 betweeen TOP and PP (Figure 4). Wang et al. (2007) showed that the mean and maximum errors of MODIS surface temperature were 0.27 K and 2.61 K, respectively at a semi-desert site on the western TP, which is comparable to the maximum difference of 1 K between TOP and PP (Figure 4). Salomonson and Appel (2004) showed that using the Landsat 30 m observations as the benchmark, the mean error of MODIS snow cover fraction was smaller than 0.1, which is comparable to the difference of 0.1 between TOP and PP (Figure 4). Mu et al. (2007) showed that the 8-day MODIS latent heat flux had a mean bias from -5.8 to 39.9 W/m$^2$, possibly larger than the difference between TOP and PP in our study (Figure 4).

Schaaf, C. B., Gao, F., Strahler, A. H., Lucht, W., Li, X., Tsang, T., Strugnell, N. C., Zhang, X., Jin, Y., Muller, J.-P., Lewis, P., Barnsley, M., Hobson, P., Disney, M., Roberts, G., Dunderdale, M., Doll, C., d'Entremont, R. P., Hu, B., Liang, S., Privette, J. L. and Roy, D.: First operational BRDF, albedo nadir reflectance products from MODIS, Remote Sensing of Environment, 83(1-2), 135–148, doi:10.1016/s0034-4257(02)00091-3, 2002.

**For question (2):**

It is difficult to directly decouple the contributions of the direct treatment of sub-grid radiation and snow cover evolution. Since there is no snow cover in summer, differences of TOP and PP are caused by the new radiation treatment (Figure 3). The absolute difference in net solar radiation can still be as large as 20 W/m$^2$ in summer, but the relative difference in summer is smaller than in winter. Generally, direct albedo of TOP shows higher consistencies with MODIS than PP, when snow cover fraction is larger or the snow cover fraction of TOP have higher consistencies with MODIS (Figure 11). These demonstrates that snow cover plays an important role in the improved processes. We added these analysis in Line 404-406 of the revised manuscript.

[Figure]

**Figure 11.** Relationship between the differences in bias for TOP and PP ($|\boldsymbol{\delta}_{TOP}|-|\boldsymbol{\delta}_{PP}|$) with respect to MODIS data for direct albedo and PP simulated snow cover fraction (a) or the differences in bias for TOP and PP ($|\boldsymbol{\delta}_{TOP}|-|\boldsymbol{\delta}_{PP}|$) for snow cover fraction (b) in winter. Red line is the regression line, and R is the correlation coefficient.

---

## Author Comment (AC3)

Dear Editors and Reviewers,

Thank you very much for your detailed and constructive comments and suggestions for our manuscript. We found the comments very helpful for improving our manuscript and have revised the manuscript accordingly.

Below are the major revisions that we have made:

1. Following the suggestion of Reviewer 1, we have reordered the subsections of Section 3 by moving the subsection describing the comparison of model simulation with MODIS data to be the last subsection.
2. We have now added more details about the algorithm theory, parameterization, model assumptions, model configurations, and inputs to the model.
3. We have expanded the discussion about sources of uncertainties from both the ELM model and remote sensing data.
4. We have replaced some subjective expressions with more accurate statements.
5. Lastly, we have reduced the number figures as suggested by Reviewer 1.

The itemized responses to the comments and suggestions from the editors and reviewers are provided below in blue font. We hope our responses address the concerns raised in the last round of review and look forward to your decision on the publication of our manuscript.

Best regards,

Dalei Hao (on the behalf of all authors)

---

## Author Response (AR2)

Dear Editors and Reviewers,

Thank you very much for your great comments and suggestions for our manuscript. We have carefully revised the grammar and typo errors throughout the manuscript. Please see below for the details.

Comments from the reviewer:

Technical corrections:

1. 333-334: "large as" -> "as large as"

   As suggested, we have revised it.

2. Fig. S5: "relation" -> "relative"

   As suggested, we have revised it.

3. 310-311: "area fraction" may be more clear than "proportion"

   As suggested, we have revised it.

4. 338: "sensible heat and heat" -> "sensible heat and latent heat"

   As suggested, we have revised it.

5. 366-367: "TOP and PP have similar or possible worse performance" do you mean TOP possibly has worse performance than PP?

   Yes, we revised it as "TOP possibly has worse performance than PP".

6. 406: "these demonstrates" -> "this demonstrates"

   We have revised it as "these demonstrate".

Thank you again!

Best,

Dalei Hao (on the behalf of all authors)